# Weight-sparse transformers have interpretable circuits

**Leo Gao** [1]   **Achyuta Rajaram** [1]   **Jacob Coxon** [1]   **Soham V. Govande** [1]   **Bowen Baker** [1]   **Dan Mossing** [1]

## Abstract

Finding human-understandable circuits in language models is a central goal of the field of mechanistic interpretability. We train models to have more understandable circuits by constraining most of their weights to be zeros, so that each neuron only has a few connections. To recover fine-grained circuits underlying each of several hand-crafted tasks, we prune the models to isolate the part responsible for the task. These circuits often contain neurons and residual channels that correspond to natural concepts, with a small number of straightforwardly interpretable connections between them. We study how these models scale and find that making weights sparser trades off capability for interpretability, and scaling model size improves the capability-interpretability frontier. However, scaling sparse models beyond tens of millions of nonzero parameters while preserving interpretability remains a challenge. In addition to training weight-sparse models *de novo*, we show preliminary results suggesting our method can also be adapted to explain existing dense models. Our work produces circuits that achieve an unprecedented level of human understandability and validates them with considerable rigor.

## 1. Introduction

While neural networks, such as large language models, have rapidly increased in capability in recent years, we still understand very little about how they work. Mechanistic interpretability seeks to reverse engineer neural networks and fully understand the algorithms they implement internally.

A major difficulty for interpreting transformers is that the activations and weights are not directly comprehensible; for example, neurons activate in unpredictable patterns that don't correspond to human-understandable concepts. One

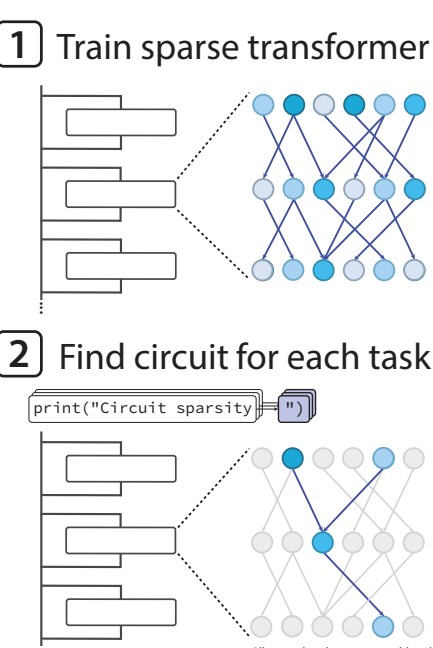

*Figure 1.* An illustration of our overall setup. We first train weight-sparse models. Then, for each of a curated suite of simple behaviors, we prune the model down to the subset of nodes required to perform the task. We ablate nodes by pruning to their mean activation value over the pretraining distribution.

hypothesized cause is superposition (Elhage et al., 2022b), the idea that dense models are an approximation to the computations of a much larger untangled sparse network.

Existing approaches have made progress on tackling superposition by first learning a basis in which activations appear sparse, and then attempting to understand the computations of the model within that basis (Marks et al., 2024; Ameisen et al., 2025). However, these approaches obtain human-understandable circuits by abstracting away complex computations that are only partially understood. Thus, the resulting circuits may reflect the chosen abstractions in addition to the model's true mechanisms.

Here, we introduce a new paradigm which leads to substantially simpler and more general circuits that we can fully understand even at the lowest levels of abstraction. To do this, we train transformers where the vast majority of weights are zeros; i.e., the $L_0$ norm of the weights is small.

[1]OpenAI, San Francisco, California, United States. Correspondence to: Leo Gao <lg@openai.com>.

*Proceedings of the $43^{rd}$ International Conference on Machine Learning*, Seoul, South Korea. PMLR 306, 2026. Copyright 2026 by the author(s).

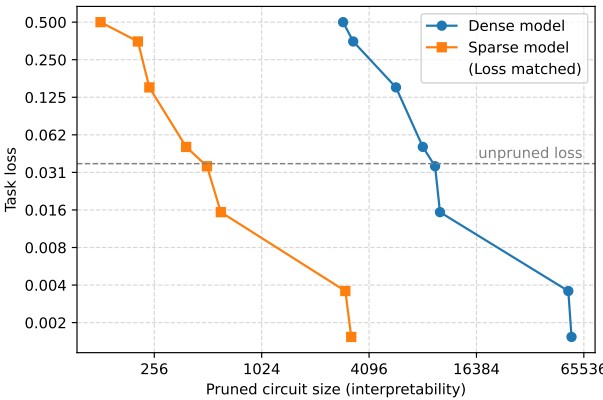

*Figure 2.* Our weight-sparse models learn simpler task-specific circuits than dense models. We examine a sparse model and a dense model with the same pretraining loss. We sweep target loss, and find the size of the minimal circuit in each model that can achieve that loss, averaged across tasks. Sparse model circuits are roughly 16-fold smaller at any given loss.

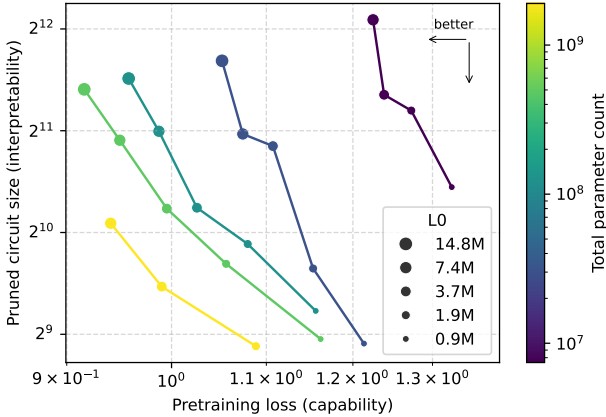

*Figure 3.* Scaling the total parameter count of weight-sparse models improves the capability-interpretability Pareto frontier. Making models sparser (*i.e.* decreasing the $L_0$ norm of weights) while holding total parameter count fixed trades off the two, harming capability but improving interpretability. We define capability as pretraining loss; see Section 2.2 for our definition of interpretability. Down and to the left is better.

This constraint drastically simplifies model computations. As each neuron can only read from or write to a few residual channels, the model is discouraged from distributing concept representations across multiple residual channels or using more neurons than strictly needed to represent a single concept.

We show that the model learns disentangled circuits for different tasks by isolating the minimal circuit which can perform each task and showing that it is compact. Within these circuits, we find that neuron activations often correspond to simple concepts, such as "tokens following a single quote" or "depth of list nesting", and the weights encode connections between concepts that are often intuitive. As a relatively rigorous validation, we further demonstrate that our disentangled circuits are necessary and sufficient for the model's behavior on these tasks; mean-ablating every neuron except the few that are part of the circuit preserves task loss, whereas deleting the few nodes in the circuit severely harms task loss.

Although weight-sparse training has substantial benefits for interpretability, it has the critical disadvantage that it requires training new models *de novo*; these models are extremely inefficient to train and deploy, and are unlikely to ever reach frontier capabilities.[1]

While we believe that training larger interpretable models is intrinsically scientifically valuable, we'd also like to use weight-sparse training to understand existing dense models. We show preliminary results using *bridges* at each layer to couple a weight-sparse model's representations to those of a target dense model, so that our model can then serve as an

interpretable replacement for the original dense model.

To make it easier to replicate our results, we release the weights and pruned circuits for all sparse models used in the experiments in this paper, as well as code for training and visualizing circuits, at https://github.com/openai/circuit_sparsity/.

## 2. Methods

We first train **weight-sparse models**—Transformer models with most of their parameters set to zero.[2] All of our models are pretrained on a dataset of Python code. We then examine our models' behavior on a curated suite of simple, unambiguous tasks where they are forced to choose between one of two completions.

To assess the interpretability of our models, we isolate the small **sparse circuits** that our models use to perform each task using a novel pruning method. Since interpretable models should be easy to untangle, individual behaviors should be implemented by compact standalone circuits.

Sparse circuits are defined as a set of nodes connected by edges. Our definition of nodes is maximally granular and corresponds to rows and columns of weight matrices: we define a **node** to mean an individual neuron, attention channel, residual channel read, or residual channel write. An **edge**, then, is a nonzero entry in a weight matrix, and connects two nodes.[3]

---

[1] See Appendix B for more discussion on the inefficiencies of sparse training.

[2] Not to be confused with sparsity via mixture-of-experts (Shazeer et al., 2017), which is weight-dense in our terminology, because its weights are almost all nonzero.

[3] For example, the simplest MLP circuit, consisting of one sin-

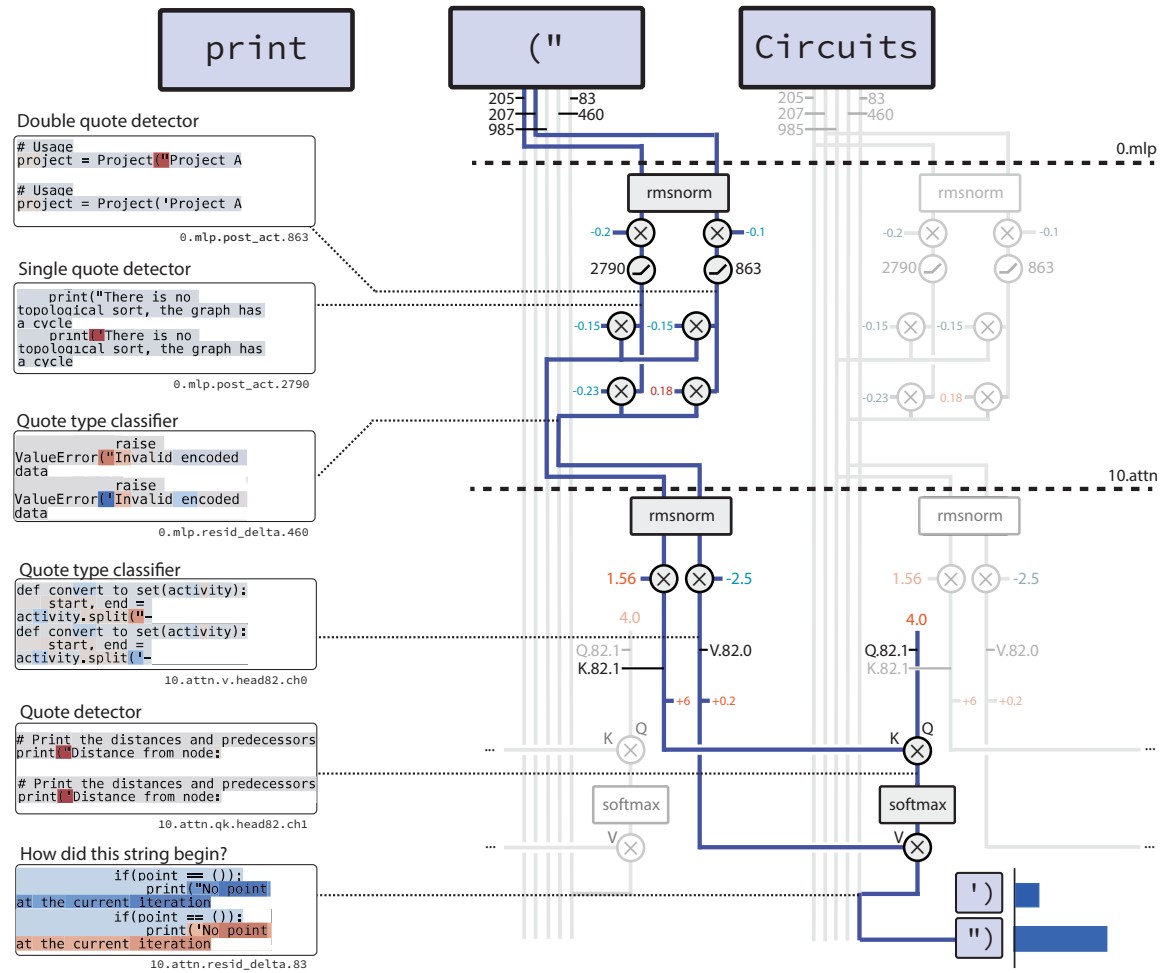

*Figure 4.* The string closing circuit. We omit no detail, showing all 12 nodes and 9 edges needed to complete the task near perfectly. First, `0.mlp` converts the token embeddings into "quote detector" and "quote type classifier" residual channels, which are read by key and value channels respectively in `10.attn`. Subsequent tokens attend to the key and copy the value to predict the corresponding closing quote. In the diagram, the vertical bundle of lines under each input token is its residual stream. Activations of important nodes on cherry-picked task examples are shown on the left. Dashed horizontal lines mark layer boundaries. ⊗ denotes scalar multiplication; directly merged lines denote scalar addition. Black numbers indicate channel or neuron indices. Red and blue numbers mark positive and negative weights (or biases). This diagram only shows the relevant attention path. Inactive parts of the circuit are greyed out, and irrelevant layers are omitted.

We report the geometric mean number of edges in the circuit across our hand-curated tasks as our main quantitative **interpretability metric**.

### 2.1. Sparse Training

**Architecture.** We use a GPT-2 style decoder-only transformer similar to Radford et al. (2019) with some minor modifications. We enforce sparsity of all weights and biases, including embeddings, such that we can increase the width of the model while holding the number of nonzero parameters ($L_0$) exactly constant. Our sparsest models have

approximately 1 in 1000 nonzero weights. We also enforce mild activation sparsity at all node locations, with 1 in 4 nonzero activations.[4] See Appendix A.1 for more details and ablations.

**Optimization.** We train to minimize cross-entropy loss using the AdamW optimizer (Loshchilov & Hutter, 2019). To enforce the $L_0$ weight sparsity constraint, we zero out all but the largest magnitude entries in each weight matrix after applying AdamW within each training step, such that every matrix has the same fraction of nonzero elements.[5] We

gle neuron reading from and writing to one channel, is 3 nodes and 2 edges. MLP neuron nodes correspond to post-GELU activations, residual read nodes correspond to the rows of `c_fc`, and residual write nodes correspond to the columns of `c_proj`.

[4]Note that this does not directly enforce sparsity of the residual stream, only of residual reads and writes.

[5]We keep gradients and Adam moments dense, not modifying them from the standard implementation.

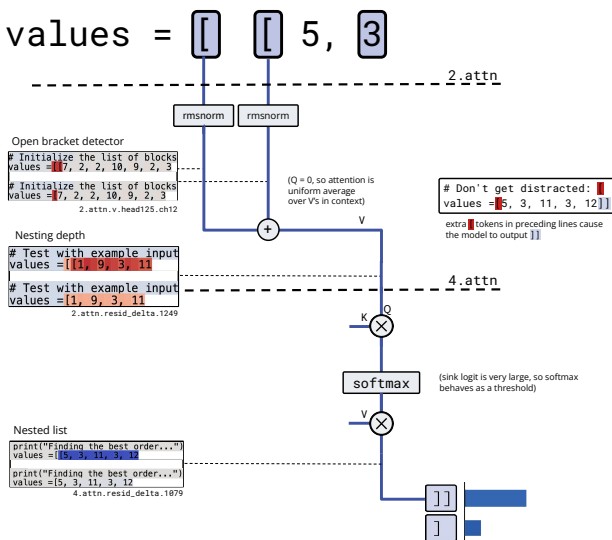

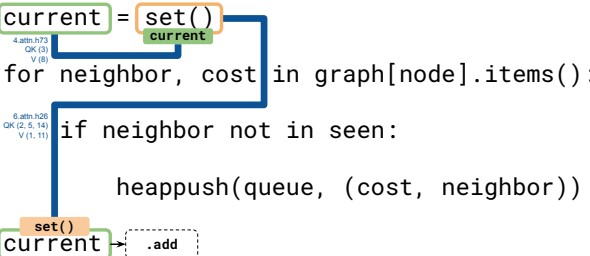

*Figure 6.* A rough schematic of the circuit for tracking the type of a variable. The model uses a two-hop algorithm with 2 attention heads, using 4 query/key channels and 3 value channels in total. First, it copies the variable name `current` into the `set()` token. It then uses this as a key, allowing the model to copy the value of the `set()` token into the final token position, where it reads out the correct answer.

*Figure 5.* A simplified illustration of the circuit for counting nesting depth, using the conventions from Figure 4. A single attention value channel functions as an "open bracket detector" derived from the embedding of the token `[`. The attention head then averages the value of this detector over the context and writes it to the residual stream at each token (the "nesting depth"). A subsequent attention head reads out the nesting depth using a query channel, and thresholds it to only activate inside nested lists. This circuit uses 7 nodes and 4 edges. Understanding this algorithm allows us to adversarially attack the model with "distractors."

anneal the $L_0$ from fully dense to the target $L_0$ throughout training. See Appendix A.2 for more details and ablations regarding techniques used to ensure optimization stability.

### 2.2. Measuring interpretability

**Task distribution.** We manually construct a set of 20 simple Python next-token binary prediction tasks. For example, one task (`single_double_quote`) is to predict whether to close a string with a single or double quote, where the only difference in the context is whether the opening quote token has a single or double quote. Another (`set_or_string`) measures the model's ability to track the type of a variable, by asking whether a variable name should be followed by `.add`, or `+=`, where examples differ only in the variable's initialization. See Table 1 for a description of all tasks.

**Pruning.** For each task, we prune the model to obtain the smallest circuit which achieves a target loss on the task distribution. The target loss is 0.15 everywhere unless otherwise specified. We prune by deleting some subset of nodes across all token positions, similar to Cao et al. (2021). Deleted nodes are *mean-ablated* — that is, their activation is frozen at the mean activation over the pretraining distribution. See Appendix E for a discussion of the implications of various ablation methods. We present a novel structured

pruning algorithm. We learn a set of masks $\tau_i$ (indexed by node) which we use to gate the respective node locations $x_i \mapsto x_i \odot \sigma(\tau_i)$, where $\sigma$ is the Heaviside step function. We train the mask parameters $\tau_i$ by using a sigmoid-derivative surrogate gradient to backpropagate through the Heaviside step function (analogous to the Straight-Through Estimator (Bengio et al., 2013)), minimizing a joint objective of task loss and circuit size. See Appendix A.5 for more details.

### 2.3. Bridges

In Section 3.3, we extend our methods to understanding behaviors of already-trained dense models. We train a weight-sparse model alongside a series of bridges mapping between the dense and sparse model's activations once per sublayer, *i.e.* before each attention and each MLP. Each bridge consists of an encoder which maps from the dense to sparse model activations, and a decoder that maps back.[6]

**Loss terms.** We want the weight-sparse model to match the dense model's computations, with the bridges accurately translating between the sparse and dense activations. To accomplish this, we use multiple bridge loss terms in addition to normal pretraining loss (Figure 7). We use an NMSE term that trains the bridge encoder to accurately predict sparse activations from dense ones (and vice versa for the bridge decoder). We also run hybrid forward passes of the sparse and dense models, using the bridges to convert between the two types of activations. We train the sparse model such that these hybrid forward passes have low KL to the original dense model. See Appendix A.3 for the precise setup.

---

[6]Each encoder is a linear map and activation function (AbsTopK), and the decoder is linear. In other words, each bridge can be viewed as a sparse autoencoder where the latent space is the sparse model's residual activations.

# 3. Results

## 3.1. Weight sparsity improves interpretability

First, we measure whether sparsity allows models to learn smaller circuits. We compute minimal circuits in a dense model and a sparse model of comparable pretraining loss for each of our tasks, and average the circuit size across tasks (Figure 2). We show that pruning our weight-sparse models yields roughly 16-fold smaller circuits on our tasks than pruning dense models of comparable pretraining loss. We are also able to construct arbitrarily accurate circuits at the cost of more edges. This shows that circuits for simple behaviors are substantially more disentangled and localizable in weight-sparse models than dense models.

To further validate the faithfulness of our circuits, we also show that they are not only sufficient but necessary for our tasks. When we ablate the tiny fraction of nodes that are part of our circuits, leaving the rest of the network intact, performance is severely impaired (Figure 32).

We also find that our method improves with model scale. When we increase $d_{model}$, holding the number of layers constant, we improve the interpretability-capability frontier (Figure 3). Changing $L_0$ moves along the frontier, trading off between capability and interpretability. Finally, if we hold $L_0$ fixed, and use a larger model, we find that both capability and interpretability improve. The larger model with the same $L_0$ is strictly more expressive, and has fewer nonzero weights per neuron or residual channel.

## 3.2. Qualitative circuit investigations

Our ultimate goal is to fully understand the computations of a model mechanistically. Our quantitative interpretability score is useful but it is still only a proxy. To verify that the model implements human-understandable algorithms, we manually interpreted the pruned circuits for three different tasks across two models (chosen for their apparent simplicity). For each circuit, we spent roughly a researcher-day of work. This included manually removing extraneous nodes, as well as verifying our natural language descriptions of nodes with manual activation patching experiments.

A unique promise of sparse circuits is that we could aspire to interpret them in the absence of any task-specific data or assumptions. Optimistically, after relating nodes to natural concepts,[7] we could extract relevant circuits by directly inspecting their edges. To this end, for the following circuits we report how numerous are the components' total edges relative to the interpreted subset, in each circuit. Generally, if components have fewer total edges, we expect that it will be easier to trace their circuits, and their edges may be more

---

[7]We believe this to be generally possible; randomly selected nodes seem to be somewhat interpretable.

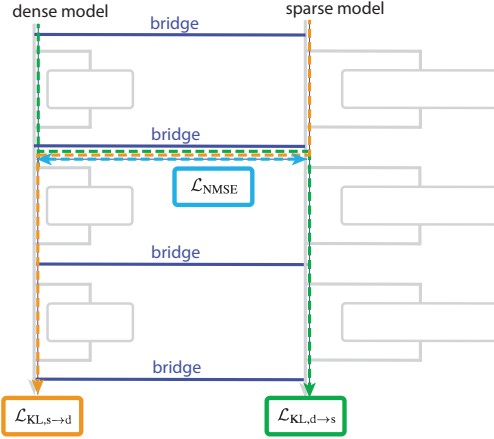

*Figure 7.* Starting with an existing dense model, we aim to train the weight-sparse model jointly with *bridges*—a series of linear maps that allow us to convert between the sparse and dense model representations—such that all paths through a mix of sparse and dense layers still perform well on pretraining.

interpretable.

### 3.2.1. CLOSING STRINGS

We begin with a simple task (`single_double_quote`), where the input contains a string opening with either a single or double quote, which must close with a quote of the corresponding type. We find that the circuit for this task uses two steps, involving two neurons in one MLP layer and one attention head (using one QK channel and one V channel). We are confident that the following mechanism reflects how the model closes strings on our task distribution.

In the first step, the earliest MLP layer (`0.mlp`) combines the embeddings for `("` and `('` into a "quote detector" neuron (channel 985, which is positive on both `("` and `('`), and a "quote type classifier" neuron (channel 460, which is positive on `("` and negative on `('`). In the second step, a layer 10 attention head (`10.attn.head82`) uses the "quote detector" as a key (channel 1), and the "quote type classifier" as a value (channel 0). As the last token has a constant positive-valued query, the attention head output successfully closes the string. A detailed schematic of this circuit is in Figure 4.

We also check whether the nodes we find have the same interpretation on the pretraining distribution. We find that some but not all nodes are exactly monosemantic even on the pretraining distribution—see Figure 39 for an example.

Together, the four components described (two MLP neurons, a QK channel, and a V channel) have 41 edges connecting them to the rest of the network, of which 9 are used by this circuit—based on this, we are optimistic it would be possible to understand this circuit without task-specific data.

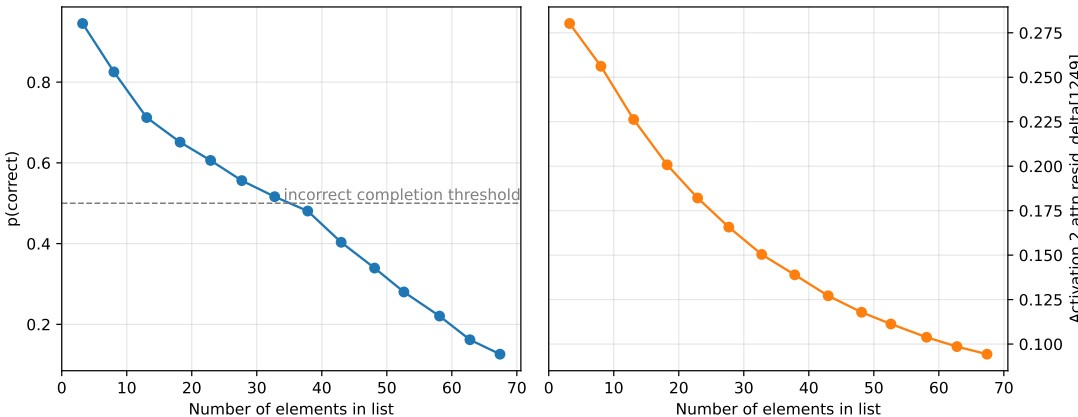

*Figure 8.* A surprising adversarial example to our bracket-counting circuit found using our circuit. Because our model uses stronger and weaker activations of the same feature to represent different bracket nesting depths, and also takes the mean of this feature over the context, it should be possible to trick the model by putting more tokens in context. We find that our model (even when unpruned) experiences significant "context dilution", failing to predict the correct completion of ` ]] ` on longer lists. This is a natural consequence of the circuit as understood in Section 3.2.2; the activation reduces in magnitude (proportional to $\frac{1}{n\_ctx}$) as the average is taken over more tokens. Strikingly, we find this attack generalizes to similarly capable dense models.

### 3.2.2. COUNTING NESTING DEPTH

Next, we examine a more complex task, probing whether a model[8] can appropriately close a flat or a nested list with ` ] ` or ` ]] `, respectively (`bracket_counting`). We manually extract a minimal circuit for this task.[9] We are reasonably confident the model counts nesting depth of lists on our task distribution with the following three steps:

**Embedding.** The token embedding of the final token is ignored. The token embedding of ` [ ` writes to residual channels 759, 826, and 1711. These channels become "bracket detectors" for the ` [ ` token.

**Counting.** Given these bracket detector activations, the model then counts the brackets using a single value channel in layer 2. The model sums the embedding channels into a value channel (head 125 channel 12) which functions as an "open bracket detector" active at each open bracket token. Attention head 125 has nearly-zero query and constant keys across the sequence, meaning that the softmax attention operation amounts to averaging over the context. Head 125 then writes the resulting averaged open bracket detector value to residual channel 1249. Residual channel 1249

---

[8]Note that this model uses attention sinks, unlike the model in the previous section.

[9]In a small number of cases, we manually remove redundant components and rescale nodes to compensate. This effectively makes use of an additional operation that our pruning algorithm doesn't have access to, which is lumping together apparently redundant nodes and treating them as a single (scaled) node. Doing this requires caution, as it can hide superposition; in this case, it seems likely to be harmless redundancy. See Appendix F.2.

encodes the "list depth" with its magnitude.

**Thresholding.** However, to determine whether we want to output ` ]] `, we need to binarize our list depth activation. This is performed by a second attention head in layer 4 (head 80) using a strong attention sink, and list depth as a query channel activation (channel 4). Tokens in flat lists have $q \cdot k \ll$ sink, while nested lists have $q \cdot k \gg$ sink. Thus, head 80 outputs a positive value to residual stream channel 1079 ("nested list close") *only* on nested lists (where the query is large enough), and outputs ` ]] `.

Our mechanistic understanding of `bracket_counting` lets us predict how the model will perform on unseen, related inputs. The circuit relies on a simple average over previous tokens, which would fail in the presence of unmatched open "distractor" brackets. Using an adversarial code comment, we can fool the model into predicting a double bracket completion on a flat list (Figure 5). Furthermore, we know that the attention circuit uses the magnitude of residual channel 1249 to represent different bracket nesting depths, and also takes the mean of this feature over the context length. It's thus possible to make the (unpruned) model incorrectly predict ` ] ` on nested lists by making the list very long, "diluting" the context. This effect correlates exactly with the activation magnitude of residual channel 1249 (see Figure 8 for more details). We find this attack generalizes to similarly capable dense models, showing that this attack is not due to idiosyncrasies of weight-sparse models.

This circuit uses 6 channels total, with 283 edges connecting them to the rest of the network. An additional 11 layer 3 attention channels and 48 layer 7 MLP neurons elided from the above contribute an additional 1217 edges. It would

likely be difficult to make progress on tracing this circuit without task-specific data.

### 3.2.3. TRACKING THE TYPE OF A VARIABLE

Even when the circuit describing a model's behavior is not perfectly disentangled into a small number of individually interpretable activations, our models tend to learn partially interpretable computation graphs. On `set_or_string_fixedvarname`, from inspecting the circuit, we believe that the model uses the following two-step algorithm for tracking whether a variable called `current` is a set or string. First, given an input `current = set()` or `current = ""`, a head in layer 4 which attends to recent tokens (head 73) copies the embedding of `current` into the `set()` or `""` token via value channel 8. Then, when the model is required to recall the value of variable "current" later in the sequence, a head in layer 6 (head 26) uses the embedding of "current" as a query and key activation. Thus, the head copies the information to complete the task from the `set()` or `""` token to the final token. This algorithm is outlined in a schematic in Figure 6.

The circuit described uses 4 QK channels and 3 V channels across two attention heads in successive layers. These channels have 100 total edges connecting them to the rest of the network. It may be difficult to understand this circuit without task-specific data.

### 3.3. Using bridges to extract circuits from existing models

So far, all of our results are from weight-sparse models trained *de novo*. However, it would be valuable if we could also use our methods to understand behaviors of already-trained models. Dense models are vastly more computationally efficient than sparse models, and it would be valuable to gain confidence that sparse models have circuits that are mechanistically analogous to the ones in dense models.[10]

We do a preliminary exploration in this direction, by training a weight-sparse model whose computations correspond with a dense model's computations at the same layer (Figure 7). Then, "interpretable" perturbations of the weight-sparse model's activations can be mapped to corresponding perturbations of the dense model's activations to achieve a desired change in behavior.

Using pruning, we identify the minimal sparse circuit that can perform well on a specific task, as described in Section 2.2. We manually choose a node from this circuit which seems to (1) be important for the task based on ablations and

(2) encode some feature of interest. We perturb these nodes in the sparse model, and linearly map the perturbation to the original dense model using bridges (see Appendix A.4).

As shown in Figure 9, this procedure allows us to construct perturbations of the dense model's activations consistent with altering a feature of interest.

In the first case, we study the `single_double_quote` task in a 4-layer dense model and bridged 4-layer sparse model. This sparse model's circuit qualitatively resembles the one described in Figure 4. We perturb a residual channel at the input to the sparse model's final attention layer which acts as a "quote type classifier" (activations shown in Figure 40). We prompt the model with a double-quoted string, and steer this channel so that its activation resembles that of a single-quoted string to construct an interpretable perturbation. After applying the bridge to this perturbation, this causes a steep increase in the dense model's probability of outputting a single quote. This is consistent with editing the model's representation of the quote type as stored in the quote token.

In the second case, we study a different task, `while_return_true`, in which the model is expected to output a `:` token after `while True` but a newline after `return True`. For this task, we examine a second bridged 4-layer sparse model coupled to the same dense model, and manipulate a channel at the input to the final MLP layer. The channel's activation is highly negative throughout lines that begin with `if`, `while`, or `except` (any of which should end in a colon; activations shown in Figure 41). We prompt the model with code ending in `return True` and steer this channel toward its counterfactual `while True` activation. After applying the bridge to this perturbation, this increases the dense model's probability of outputting a colon rather than a newline (though not as steeply as the previous task). This is consistent with partially editing the dense model's representation of whether the current line should end in a colon based on the first token.

## 4. Discussion

Due to fundamental constraints, unstructured weight-sparse neural networks are unlikely to ever approach the efficiency of dense networks (see Appendix B). Therefore, it will be infeasible to use our method to fully interpret frontier models, or to train interpretable frontier models *de novo*. We need to overcome this barrier to help us improve our understanding of frontier models. There are two main avenues that we're excited about.

First, we could scale our methods to create interpretable model organisms up to the scale of GPT-3. It seems plau-

---

[10]As some evidence for this, we find that the tokens that are hard/easy for dense models are largely the same as the ones that are hard/easy for our weight-sparse models (Figure 33).

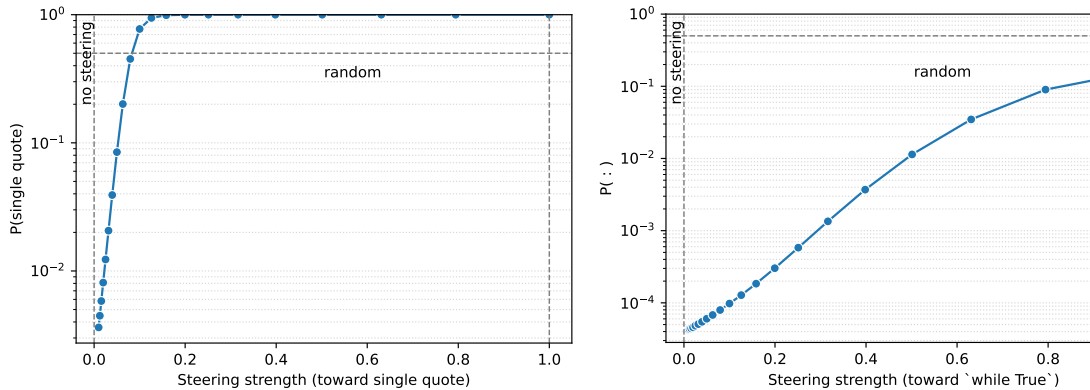

*Figure 9.* Using sparse models and bridges to edit representations of an existing dense model. We perform "interpretable perturbations" on dense model activations in two tasks (`single_double_quote` and `while_return_true`) to test whether a bridged sparse model is faithful to a dense model. On the left, we attempt to edit the dense model's "quote type classifier" representation to induce the model to behave as if it were prompted to complete a single- rather than a double-quoted string using bridges. On the right, we attempt to edit the dense model's representation of whether the current line began with `if`, `while`, or `except`, to induce the model to behave as if it were prompted with `return True` rather than `while True`. The dense model's behavior is consistent with at least partially successful editing in both cases.

sible that transformers learn universal circuit motifs that appear in both sparse and dense models across scales. If so, then studying the circuit motifs of these model organisms would give us a sense of what motifs to look for in frontier models and help us better target our investigations.

In particular, if we created model organisms whose computations were coupled to dense models via bridges, comparing their computations could be valuable for studying phenomena such as superposition and interference weights in dense models (Olah et al., 2025).

Second, although understanding a frontier model across its entire pretraining distribution is prohibitively expensive, we could save on compute by seeking to understand less. In particular, we could train a sparse bridged model on a narrow but important task distribution (*e.g.*, deception, refusal, goal-seeking). Although this might not enable ambitious reverse engineering of a frontier model's behavior, this could be a valuable tool for safety cases.

We are also excited about using sparse circuits to support automated interpretability. Sparse circuits, like dictionary learning approaches, provide new primitives for understanding model computations—a new language in which computations are simpler to express. We suspect that automated interpretability is bottlenecked on these kinds of primitives, making sparse circuits a natural complement to automation.

## 5. Limitations and Future Work

**Compute (in)efficiency.** Sparse models require vastly (100–1000x) more training and inference compute than dense models of comparable capability. There is considerable room for improvements to both optimization and systems. For example, our current procedure leads to many dead neurons and likely does not maximally efficiently explore different sparsity masks. There is also room for systems improvements (see Appendix B for more discussion).

**Polysemantic features.** Our circuits, especially for more complex tasks, do not consist entirely of monosemantic nodes or edges; often concepts are still smeared across a number of nodes which also perform other tasks (albeit to a much lesser extent than dense models). This could indicate that superposition is fundamentally advantageous; it may also be that scaling $d_{\text{model}}$ to be nearer to the size of typical SAEs (whose hidden dimensions are in the millions rather than in the thousands) would resolve this.

**Non-binary-valued features.** Our features are not all binarizable, meaning that they sometimes carry information in their magnitude beyond whether they are on or off. If a feature can't be discretized, it means that to fully explain it, we need to explain not just its activity pattern but also its magnitude. See Appendix C for more discussion.

**Defining faithfulness.** Mean ablation is not perfect. Ultimately, some variant of causal scrubbing (Chan et al., 2022) is necessary to gain full confidence in the faithfulness of our circuits. See Appendix E for more discussion.

**Defining interpretability.** The notion of "interpretability" we use (having compact task-specific circuits) does not fully capture intuitive notions of interpretability. Our qual-

itative investigations point at a stronger notion of human-understandability, which we could attempt to codify in an improved interpretability metric.

**Better pruning.** Our pruning method prunes nodes due to computational simplicity, but it would be ideal to prune edges directly.

**Scaling beyond small models and simple tasks.** We have started by explaining narrowly-trained models on simple tasks, and it is uncertain how our method would scale. Even under optimistic assumptions, our technique would produce extremely large and complicated circuits when explaining complex behaviors in more capable models. These circuits may require AI assistance to understand, or perhaps even fundamentally defy simple description.

## 6. Related Work

**SAEs and circuits.** In recent years, many works have explored and refined the use of SAEs in finding interpretable concepts in transformer language models (Sharkey et al., 2022; Bricken et al., 2023; Templeton et al., 2024; Gao et al., 2024; Rajamanoharan et al., 2024; Lindsey et al., 2025). Olah et al. (2017) first found circuits in image models. A number of works have also attempted to find circuits using SAEs (Marks et al., 2025; Ameisen et al., 2025; Lai, 2025), Transcoders (Dunefsky et al., 2024), and using dense models directly (Wang et al., 2022; Conmy et al., 2023; Lieberum et al., 2023). Elhage et al. (2022a) trains interpretable-by-design models with activation sparsity using a softmax activation function. Braun et al. (2024) trains SAEs end-to-end on downstream KL.

Conmy et al. (2023) creates circuits out of coarse units such as entire attention heads. Cao et al. (2021) applies a gradient-based pruning algorithm to dense models to find circuits, but such circuits are too large to be directly interpreted. Marks et al. (2025) builds circuits out of SAE features, but can only explain model behavior with circuits of thousands of nodes and hundreds of thousands of edges. Ameisen et al. (2025) yields circuits on production models by using linear "attribution graphs" between cross-layer transcoder features. However, it fails to fully explain attention patterns, and suffers from noisy "global weights" between features, such that per-prompt attribution is required to get clean circuits. Kamath et al. (2025) builds on this work and makes progress on explaining attention patterns by computing attribution scores through query-key interactions; but the dense "head loadings" do not usually result in simple attention circuits due to superposition across heads.

**Alternatives to activation sparsity.** A number of works have explored arguments for constraints other than acti-

vation sparsity (Hänni et al., 2024; Chughtai & Bushnaq, 2025). APD (Braun et al., 2025) and SPD (Bushnaq et al., 2025) decompose weights into a sum of simple components, attempting to sparsify causal attributions. Farnik et al. (2025) encourages sparsity of computation by training SAEs such that the Jacobian of features is sparse. Wong et al. (2021) trains the final layer to have sparse weights for interpretability. Friedman et al. (2023) learns RASP-like (Weiss et al., 2021) circuits.

**Sparse weight training.** Sparse training is a problem that is well-studied in the literature (Mocanu et al., 2018; Lee et al., 2019; Evci et al., 2020; Louizos et al., 2018; Dettmers & Zettlemoyer, 2019). Evci et al. (2021) is similar to our method but uses a slightly different drop criterion. Jayakumar et al. (2021) retains a full set of dense weights on the CPU, rather than zeroing out weights outside the top-$k$. Zhu & Gupta (2017) trains weight-sparse models by annealing from an initial dense model.

**Pruning.** Pruning is also a well studied problem (Han et al., 2016; Frankle & Carbin, 2019; Blalock et al., 2020; Frantar & Alistarh, 2023; Sun et al., 2024). The classic second order approaches to pruning (LeCun et al., 1990; Hassibi & Stork, 1993; Frantar et al., 2023) are expensive to compute or rely on approximations. Attribution patching (Syed et al., 2023; Kramár et al., 2024) uses a first order Bhaskar et al. (2025) prunes circuits for edges rather than nodes. Michaud et al. (2025) explores pruning for localizing model skills. Conmy et al. (2023) applied iterative structured pruning to minimize a similar task loss objective. Cao et al. (2021) learns fine-grained masks via gradient descent, with a smooth estimator of the step function.

## Acknowledgements

We thank Joshua Batson, Trenton Bricken, Lucius Bushnaq, Will Depue, Tom Dupré la Tour, Elias Frantar, Gabriel Goh, Johannes Heidecke, Jack Lindsey, Samuel Marks, Jake Mendel, Alex Makelov, Eric Michaud, Neel Nanda, Chris Olah, Jakub Pachocki, Asher Parker-Sartori, Alec Radford, Logan Riggs, Shibani Santurkar, Lee Sharkey, Rajan Troll, Jeff Wu, Yolanda Xie, and Rowechen Zhong for valuable discussions and feedback. We thank Ryan Kaufman for assistance with task creation.

## Impact Statement

This paper presents work whose goal is to advance the field of machine learning. There are many potential societal consequences of our work, none of which we feel must be specifically highlighted here."

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

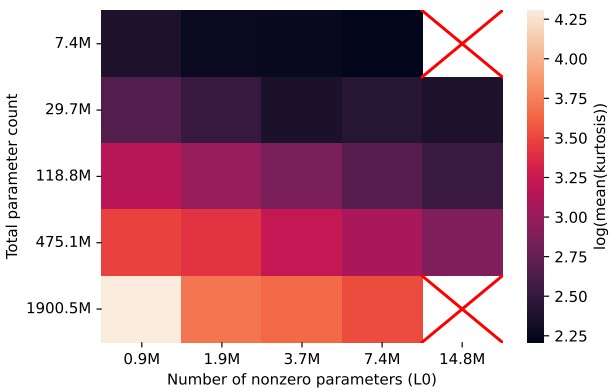

*Figure 10.* We find that weight sparsity induces activation sparsity in the residual stream. Since the residual stream entries are rarely exactly zero, we measure kurtosis instead. As the weight $L_0$ gets smaller, or the number of total parameters gets larger, the kurtosis of the final residual stream activation increases.

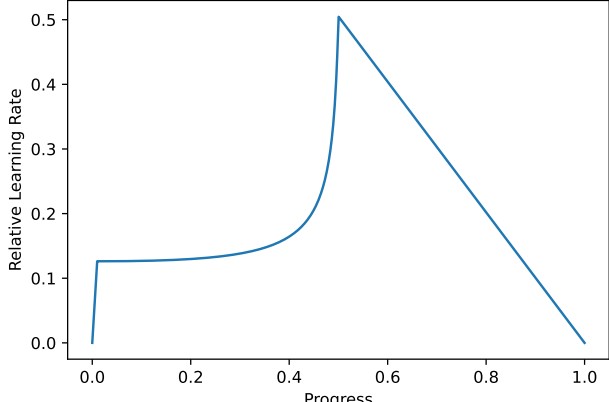

*Figure 11.* The sharkfin lr schedule with 1% warmup and $L_0$ decay for the first 50% of training.

## A. Method details

Many of our experiments were run with slightly different settings at different points in time. As a result, numbers across different plots are difficult to compare, and these method details are a general rule, but not exactly applicable to every model we train.

### A.1. Architecture

Our architecture is very close to a standard GPT-2 Transformer architecture (Radford et al., 2019). Most experiments use $n_{\text{layer}} = 8$, $d_{\text{model}} = 2048$, $n_{\text{ctx}} = 256$, unless stated otherwise. To ensure that zero values have a privileged meaning in the residual stream, we use RMSNorm (Zhang & Sennrich, 2019) instead of LayerNorm. Using RMSNorm also enables us to fold all normalization weights into the MLP/attention weights without altering the weight $L_0$. Our embedding and unembedding matrices are untied. For some models, we use attention sinks (a per-head learnable attention denominator bias) (Xiao et al., 2023), which we find leads to cleaner attention circuits, without impacting loss substantially (Figure 18).

To enforce small $L_0$ norm of activations, we apply an AbsTopK activation function at various locations in the model, zeroing out all but the $k$ largest values by magnitude at various locations; we generally set $k$ to $\frac{1}{4}$ of the dimension of the location, unless otherwise indicated. See Figure 12 for an illustration of where the AbsTopK activation functions are placed for attention and MLP layers respectively. We find that some amount of activation sparsity helps on top of weight sparsity, but that too much activation sparsity hurts the capabilities-interpretability frontier again (Figure 37). We also have some evidence that activation sparsity emerges naturally from weight sparsity: we observe that kurtosis of

activations increases with increasing weight sparsity (Figure 10).

We also add a bigram table—a single dense $d_{\text{vocab}} \times d_{\text{vocab}}$ matrix, with the most recent token's entries added to the final logits—to avoid the need for the sparse parameters to memorize bigram frequencies. Intuitively, it is desirable to add dense components to the model that have very simple interpretations (prior log probabilities over bigrams); they help improve the loss (Figure 24), and avoid valuable MLP and attention space being taken up by this information.

In most experiments we do not use positional embeddings at all (Haviv et al., 2022); we find that this is roughly neutral on loss (Figure 26). In some earlier experiments we used learned absolute positional encodings, concatenated to the residual stream as read-only channels: we find that this improves interpretability, leading to sparser activation patterns within the attention heads.

We use small $d_{\text{head}}$ (16 for most experiments in this paper), which we find anecdotally to improve monosemanticity of attention heads, at the cost of systems efficiency.

### A.2. Weight-Sparse Model Optimization

We use AdamW (Loshchilov & Hutter, 2019; Kingma & Ba, 2014) with $\beta_1 = 0.9, \beta_2 = 0.95, \lambda = 0.1, \epsilon = 0.1$, and sweep lr for every experiment. See Figure 27, Figure 28, Figure 29, and Figure 30 for ablations of these hyperparameters.

We also avoid zeroing values that would cause a neuron or attention channel to have fewer than $j = 4$ nonzero values, to reduce the chance of dead neurons (Figure 22).[11]

---

[11] It hurts the loss, and has an unclear impact on interpretability. We're not sure if we would include it in future runs. We also think this could reduce the apparent qualitative monosemanticity of randomly sampled nodes; it potentially artificially keeps some

| TASK NAME | DESCRIPTION |
|---|---|
| `single_double_quote` | Predict whether a string should be closed with `')` or `")` |
| `final_kwarg` | Predict whether a function argument has a default in the definition (because previous arguments were kwargs), by measuring probability of `=None` vs `):\n`. |
| `set_or_string` | Track whether a variable is a `set()` or string, and predict whether to complete the input with `.add` or `+=` (these particular types chosen for tokenization reasons) |
| `set_or_string_fixedvarname` | Same as `set_or_string` but the variable name is always the same. |
| `class_init` | Given a class init with many lines of the form `self.x = x`, learn to copy the variable name after the equals sign. |
| `with_as` | differentiate between using with as statements and variable declarations for file loading (the former requires the model to predict `as`) |
| `bracket_brace` | Predict whether the next token should be a `:` or a `,` based on whether we're currently in a dict or a list. |
| `with_open` | predict whether to read `'r'` or write `'w'` to a file in a python open statement based off of its name. |
| `bracket_counting` | Predict `]` or `]]` depending on how many layers of bracket nesting we're in. |
| `for_while` | Predict `in` or `:` based on if a loop in python is a for loop or a while loop. |
| `else_elif` | Predict `:` or not `:` based on if a else statement is else or elif. |
| `fstring_brace` | Predict whether a print statement involves an fstring or not (by whether or not a model inserts a variable name in the print). |
| `if_ternary` | Predict `:` depending on whether or not an if statement is inside a ternary expression. |
| `var_swap` | Predict `x` when asked to swap variables x and y using x, y = y, `x` |
| `indent_for` | Predict the correct indentation following a for loop in python. |
| `lambda_func` | distinguish lambda expressions from functions, predicting `:` for lambdas (x = lambda func `:` ), and ( for functions, def func `(` |
| `if_equals` | Predict x `==` or x `=` depending on if the statement is an if conditional or variable declaration. |
| `enumerate_range` | Predict `enumerate` vs `range` depending on whether a for loop in python is over one or two variables |
| `var_if` | Predict if x == True `:newline` or x = True `newline` depending on if the statement is an if conditional or variable declaration. |
| `while_return_true` | Predict the correct indentation following a while true or return true statement. |

*Table 1.* List of all handcrafted tasks we created.

We anneal the $L_0$ linearly over the first 50%[12] of training (Figure 17), so that the model becomes sparser throughout training (Zhu & Gupta, 2017). Our lr schedule is defined by the product of a normal warmup-decay schedule, and a factor of $1/\sqrt{L_O}$, as we find smaller $L_0$ requires larger lrs. We find that this change is necessary to get any benefit from $L_0$ annealing (Figure 25).

We clip the root-mean-square of the gradient to 1. We find gradient clipping essential for ensuring training stability (Figure 16).

At some point, we identified a bug in our $L_0$ scheduling that meant while our weights had the intended $L_0$ schedule, the embedding, unembedding, and biases would go from

dense to sparse in a relatively small number of steps in the middle of training. However, fixing this bug seems to very slightly hurt model quality in terms of both capabilities and interpretability, so we kept it in. (Figure 21).

We find that lr warmup (for the first 1% of training in most of our experiments) is critical for stability at higher lrs, substantially improving optimal loss (Figure 13). We also find that doing a longer warmup is slightly beneficial (Figure 14).

We generally don't find improvements of the capability-interpretability pareto frontier from increased token budget on the current margin; while increasing token budget improves the pretraining loss, it generally hurts the pruned circuit size.

---

irrelevant neurons alive that "want" to be dead.

[12]For our largest, sparsest models, we found it beneficial to increase this to 80%.

### A.3. Bridges loss terms

We want the weight-sparse model to match the dense model's computations, with the bridges accurately translating between the sparse and dense activations. To accomplish this, we use three bridge loss terms in addition to normal pretraining loss. Let $h_i^d$ and $h_i^s$ be the residual activations of the respective models at layer $i$, $M_i^d$, $M_i^s$ be the sublayers of the model (either an MLP or attention block), so that $h_0^d$, $h_0^s$ are the token embeddings of the dense and sparse models, and $M_i^d(h_i^d) = h_{i+1}^d$, $M_i^s(h_i^s) = h_{i+1}^s$), $f_i$ and $g_i$ be the bridge encoders and decoders, and $y^d = M_{\text{unemb}}^d(h_L^d)$ as the final logits. First, we have a normalized MSE term

$$\mathcal{L}_{\text{NMSE}} = \sum_i^L \text{NMSE}(f_i(h_i^d), h_i^s) + \text{NMSE}(g_i(h_i^s), h_i^d)$$

Second, we have a KL term to train the sparse model to accept activations from the dense models

$$\mathcal{L}_{\text{KL,d}\to\text{s}} = \sum_i \text{KL}\big(y^d, (M_{\text{unemb}}^s \circ M_L^s \circ \cdots \circ M_i^s \circ f_i)(h_i^d)\big)$$

and another term for the reverse—dense models accepting activations from sparse models

$$\mathcal{L}_{\text{KL,s}\to\text{d}} = \sum_i \text{KL}\big(y^d, (M_{\text{unemb}}^d \circ M_L^d \circ \cdots \circ M_i^d \circ g_i)(h_i^s)\big)$$

(Figure 7).

Ideally, we'd like to train over all $2^L$ possible combinations of sparse and dense layers. We can think of our KL terms as the first order approximation of this, where we consider only the subset of combinations with one transition between sparse and dense.

### A.4. Bridge intervention

Bridges are trained to convert between residual stream locations in a dense and in a sparse model. We would like to reuse these bridges to convert single-node interventions in sparse models to "interpretable" dense interventions in dense models.

Bridges act on residual stream locations (pre-RMSNorm), and so they learn to compensate for differences in scale of the residual stream between sparse and dense models. Because these locations are not "nodes" in our framework, when performing interventions we instead use the next nodes that exist, which are post-RMSNorm residual stream reads. The residual stream scale consideration does not apply post-RMSNorm, where both models have activations with $\sqrt{d_{model}}$-scale norm. To compensate, we multiply the

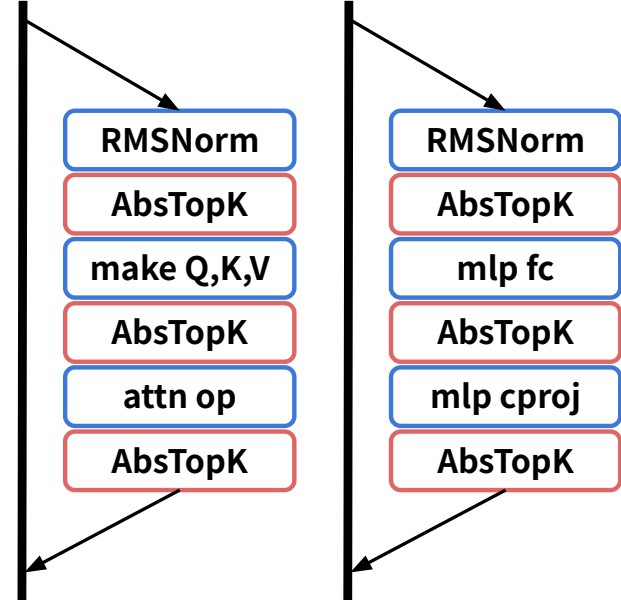

*Figure 12.* An illustration of our attention and MLP blocks, respectively. The AbsTopK activation functions are inserted at each node position, between each operation. AbsTopK is applied to Q,K, and V seperately.

bridge weights by the ratio of RMS residual stream activations averaged over a reference dataset before performing the intervention.

To construct the sparse model intervention, we first subtract the activation of the channel of interest in the presented condition (e.g. double quote) from the activation in the counterfactual condition (e.g. single quote) at all tokens. We then take the outer product with the corresponding row of the bridge (scaled as described above) and construct a tensor of tokens by dense model hidden dimension. We scale this by a "steering strength" between 0.0 (no intervention) and 1.0 (fully patched).

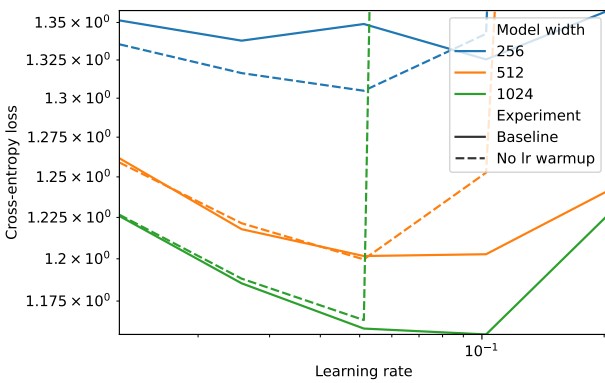

*Figure 13.* Learning rate warmup ablation

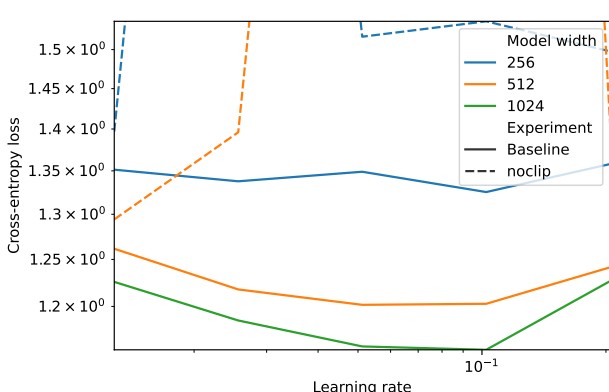

*Figure 16.* Grad clipping ablations

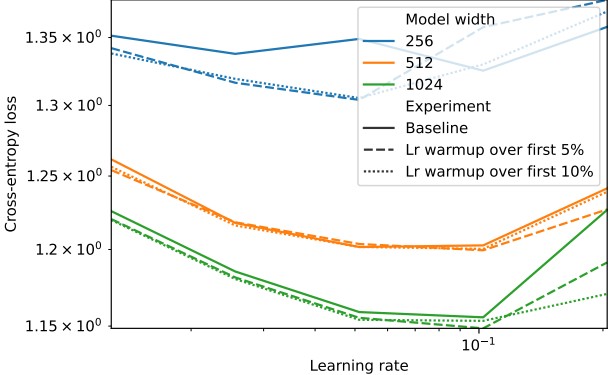

*Figure 14.* Learning rate warmup fraction ablation (1%)

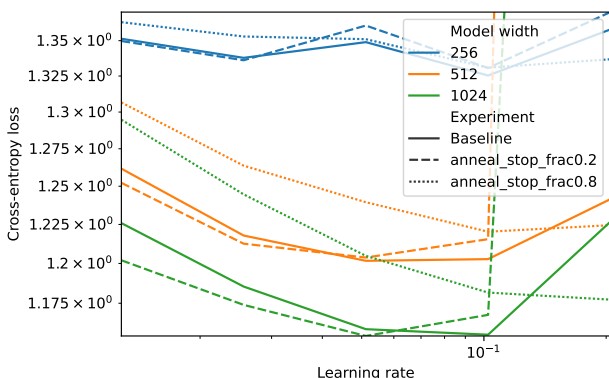

*Figure 17.* Fraction of training spent annealing ablation (baseline is 50%)

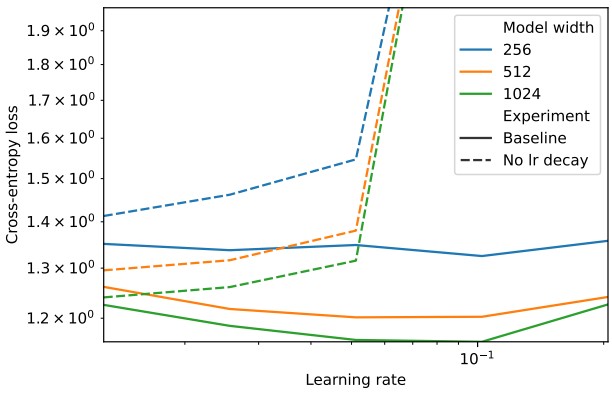

*Figure 15.* Learning rate decay ablation

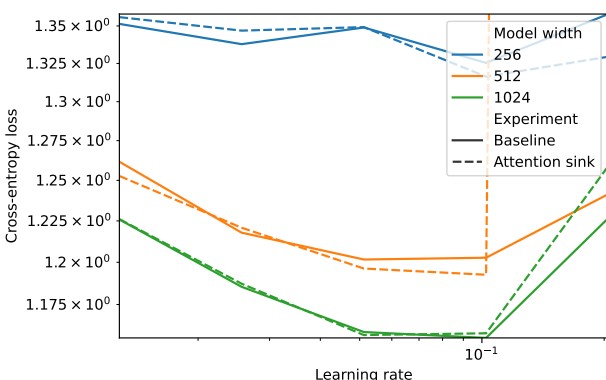

*Figure 18.* Attention sink ablation

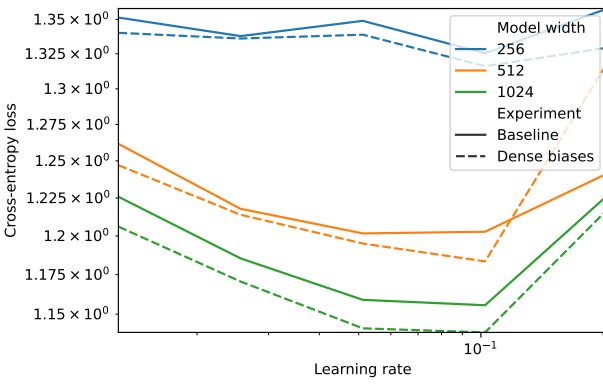

*Figure 19.* Sparse bias (baseline) vs dense bias ablation

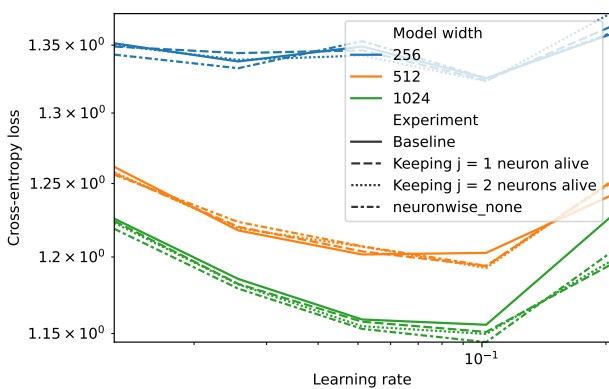

*Figure 22.* Number of forced-alive weights per neuron ablation (baseline is $j = 4$)

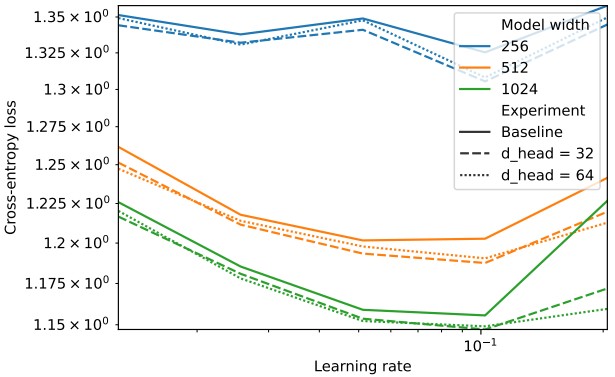

*Figure 20.* d_head sweep (baseline = 16)

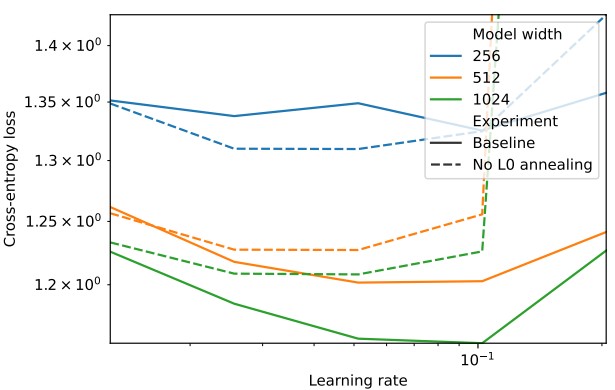

*Figure 23.* $L_0$ annealing ablation

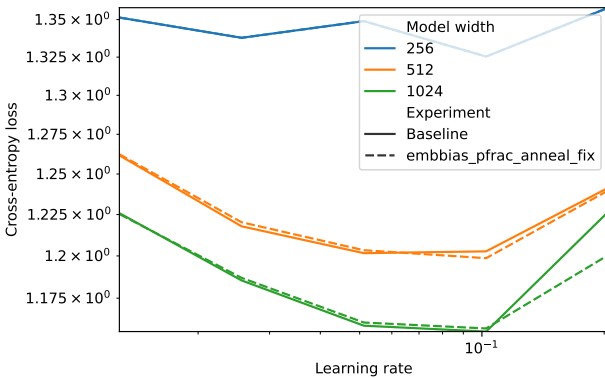

*Figure 21.* $L_0$ decay schedule for biases and embedding matrices

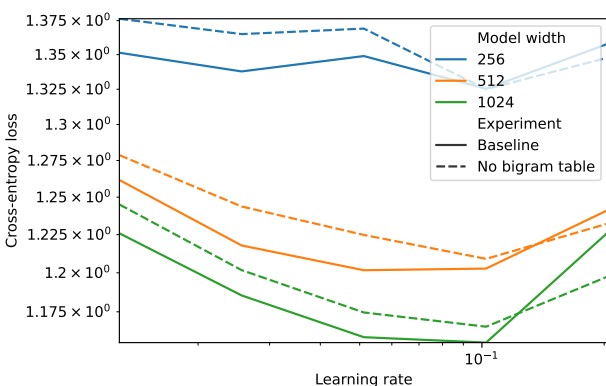

*Figure 24.* Bigram table ablation

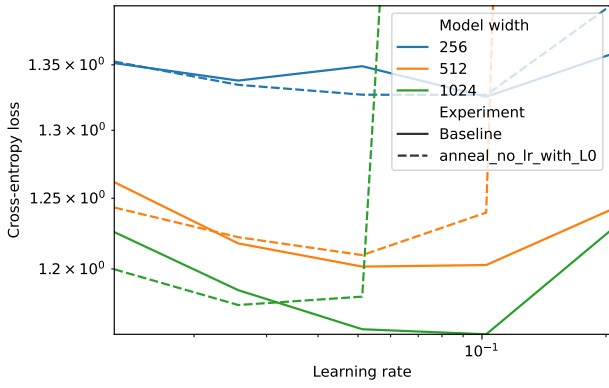

*Figure 25.* Sharkfin (baseline) vs normal warmup-decay lr schedule

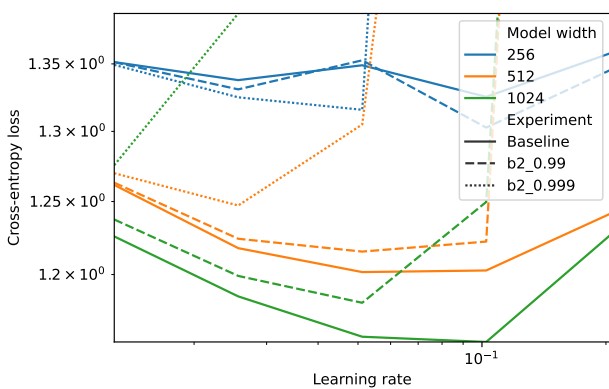

*Figure 28.* Adam $\beta_2$ ablation (baseline = 0.95)

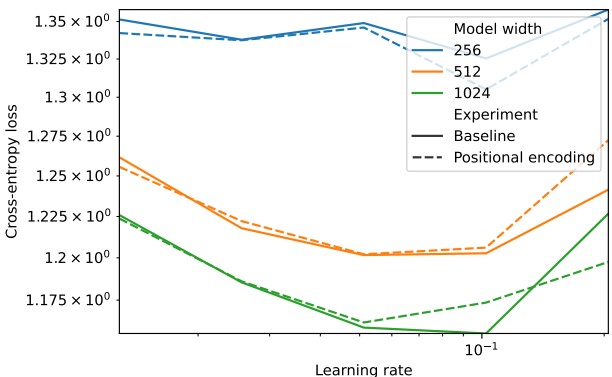

*Figure 26.* Positional embedding ablation

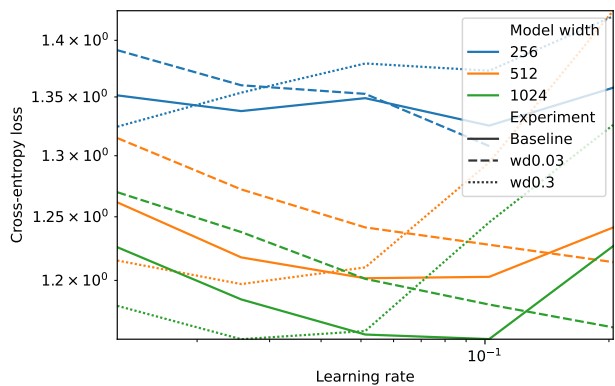

*Figure 29.* Adam $\lambda$ ablation (baseline = 0.1)

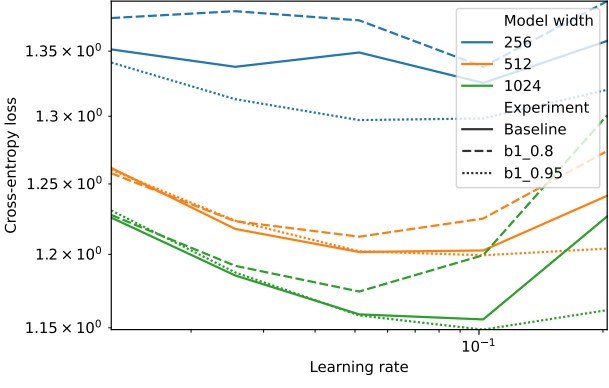

*Figure 27.* Adam $\beta_1$ ablation (baseline = 0.9)

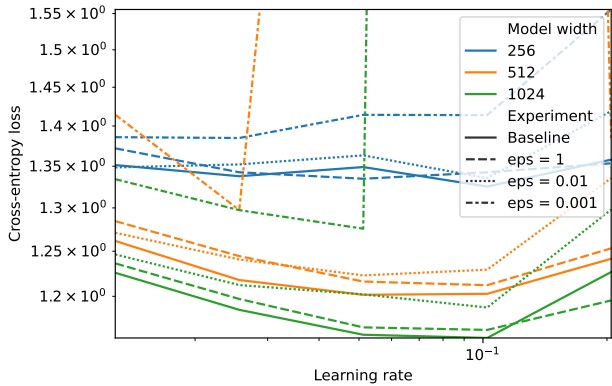

*Figure 30.* Adam $\epsilon$ ablation (baseline = 0.1)

## A.5. Pruning algorithm

**Setup.** First, we insert boolean masks in several locations in the model:

- Immediately after each RMSNorm in the attention and MLP blocks

- At the very end of each attention and MLP block, right before adding the result back to the residual stream

- After each MLP activation

- After the attention q, k, v activations[13]

We consider each element of the mask in each of these locations to be a "node", and the mask determines whether the node is included or excluded from the circuit. We reuse this definition of node in many other parts of this paper. Note that we apply the mask at the level of nodes, and not edges (e.g. interactions between nodes) as some previous pruning approaches (Bhaskar et al., 2025). We apply the mask uniformly across all prompts and across all tokens.

We learn a parameter for each node that is clamped to $[-1, 1]$, which is used to compute a boolean mask at each step by passing the parameter through a Heaviside step function, which determines which nodes are included in the resulting circuit. By optimizing the mask, we can learn extremely small circuits (compared to baselines like selecting the top nodes by gradient attribution to task loss Figure 31).

**Initialization.** We initialize with Gaussian noise scaled by a factor of `init_noise_scale`, and centered at `init_noise_bias`, and clamp the mask to $[-1, 1]$. We enforce this clamping constraint after every training step.

**Optimization.** We optimize the masks using AdamW with grad clipping. We linearly decay lr through training without any warmup. To perform backpropagation through the Heaviside step function, we use a sigmoid estimator to compute a biased gradient approximation in the backward pass, with temperature `heaviside_temp`.

**Loss function.** We optimize our masks on a linear combination of task cross entropy and $k$, the number of nonzero elements in the mask. We weight these losses via a term `k_coef`.

**Mask discretization.** After training, we bisect for the $k$ that exactly achieves the threshold; we've observed that

| Hyperparameter | CARBS search_center |
|---|---|
| `k_coef` | $1 \times 10^{-4}$ |
| `init_noise_scale` | $1 \times 10^{-2}$ |
| `init_noise_bias` | $1 \times 10^{-1}$ |
| `wd` | $1 \times 10^{-3}$ |
| `lr` | $3 \times 10^{-3}$ |
| `inv_beta2` | $5 \times 10^{-2}$ |
| `lr_warmup_frac` | $5 \times 10^{-2}$ |
| `heaviside_temp` | $1 \times 10^{0}$ |

*Table 2.* Initial search centers for CARBS

sometimes this diverges substantially from the final k output by the training procedure. As we find that our discretized models often are quite uncalibrated, we optimize a scale+shift transformation to the final logits using 16 steps of LBFGS (Liu & Nocedal, 1989). It's unclear whether this is principled to do in general.

**Hyperparameter optimization.** Because of the large number of hyperparameters, and the difficulty of setting them correctly, we use CARBS (Fetterman et al., 2023) to retune the hyperparameters for each combination of model and task. We run 32 iterations of CARBS with 8 parallel pruning jobs per iteration[14], starting from the initial hyperparameters given in Table 2. We generally find that results are poor on the first iterations and improve dramatically towards the end of CARBS tuning.

We use a batch size of 64 task datapoints (each consisting of a positive and negative sequence), for 128 total sequences of length up to 256 tokens.

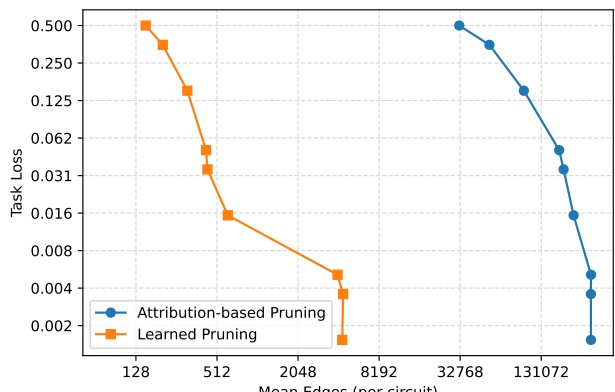

*Figure 31.* Learned pruning drastically outperforms a baseline of attribution-based pruning, finding much smaller circuits at all loss targets.

---

[13]Since q and k channels are paired most of the time, this means that each attention qk pair generally uses up two nodes. In retrospect, queries and keys should have used tied masks.

[14]That is, we do 256 total steps of the CARBS optimizer, but we alternate generating 8 suggestions, running all of them, updating on those 8 results, and repeating.

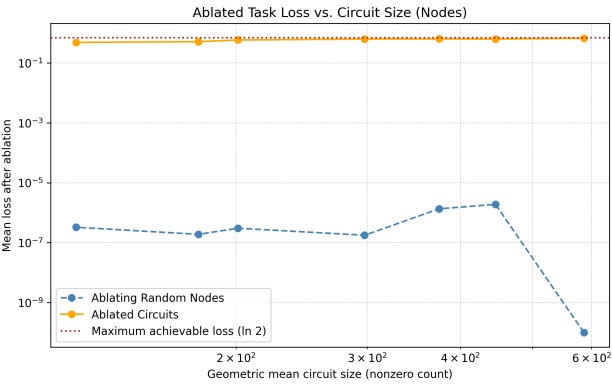

*Figure 32.* Inverse Pruning: ablating the circuit found by pruning cripples loss of the overall model. As the circuit found increases in size (and task performance), the model becomes (slightly) worse without it.

### A.6. Dataset

Our data consists of Python code generated from a mix of GPT-4-base and GPT-4o-base. The data consists of two components: a 10 billion token component of Python code designed to consist of especially simple and repetitive idioms, and a 25 billion token component designed to be closer to the full distribution of Python code.

Our decision to use a simpler data component was inspired by Eldan & Li (2023), and our hope is that our mix makes it easier to observe interesting circuits at smaller scale without making the data too toy, though we did not ablate this. Some of the smaller scale experiments in this paper are trained only on the simpler component.

We train a 2048 token BPE tokenizer on our dataset.

### A.7. Task tokenization considerations

For `single_double_quote`, we consider the tokens `("` and `('`, since they always appear as their own tokens, unlike `"` and `'`, which might be tokenized in diverse ways. Likewise for the closing quote tokens.

For `bracket_counting`, `],` is one token, so `]` is unlikely to indicate end of a non-final row.

## B. Systems considerations for scaling

Our weight-sparse training and pruning procedures optimize for interpretability, not for hardware efficiency. In particular, the (per-tensor) top-$k$ magnitude masking used to enforce $L_0$ constraints induces *unstructured* sparsity. From a systems perspective, unstructured sparsity is unfriendly to modern GPUs: gathers/scatters break the tiled dataflow that high-performance GEMM kernels rely on, reduce data reuse from on-chip SRAM, and typically prevent the use of

Tensor Cores. Since Tensor Cores provide the vast majority of available math throughput,[15] the default dense pathway is almost always preferable to attempting to implement matrix multiplies on CUDA cores.

**GEMM dataflow on Hopper and Blackwell Architectures.** Modern high-performance GEMM kernels typically use a tiled pipeline designed around bulk movement of *dense* submatrices and compute on Tensor Cores:

1. **Tile movement via Tensor Memory Accelerator (TMA).** The TMA initiates asynchronous transfers of contiguous tensor tiles from global (HBM) to shared memory (SMEM).

2. **MMA on tiles.** Tensor core instructions operate on large tiles of data in SMEM and compute matrix multiplies of up to size 128x256.

3. **Epilogue and global store.** Accumulators are optionally fused with bias/activation epilogues and written back to HBM.

This dataflow assumes regular, contiguous tiles. Unstructured masks (our top-$k$ selection) create irregular sparsity patterns that defeat bulk TMA copies and steady tensor core instruction issues.

Even on non-GPU hardware, one can expect sparse models to be significantly inefficient compared to dense ones, due to the fundamental complexity required to implement sparse GEMMs. Using a tiled dataflow with tile size greater than one would result in the memory bandwidth being wasted on moving mostly zeros, so one needs to directly route the weights individually. On a fundamental level, one needs "extra space" on the GPU die to wire the any-to-any connections (to move each entry in each weight matrix from memory to its corresponding hardware arthimetic circuit), so even a hardware implementation heavily optimized for sparse models cannot achieve the same compute density as those using systolic array-like architectures. This limitation means that it is impossible for weight-sparse models as understood here to catch up to dense models in terms of kernel efficiency. Nonetheless, we present some ideas to improve efficiency on modern GPUs above the baseline naive implementation.

**Option 1: CUDA Core GEMMs.** Depending on the ratio between the CUDA core FLOPS and the Tensor Core FLOPS on the accelerator, we can determine to dispatch to

---

[15]On H100: ∼989 TFLOPs of half-precision matrix-multiply throughput on Tensor Cores versus ∼60 TFLOPs for "everything else," i.e., CUDA cores and scalar units; see https://hazyresearch.stanford.edu/blog/2024-05-12-tk.

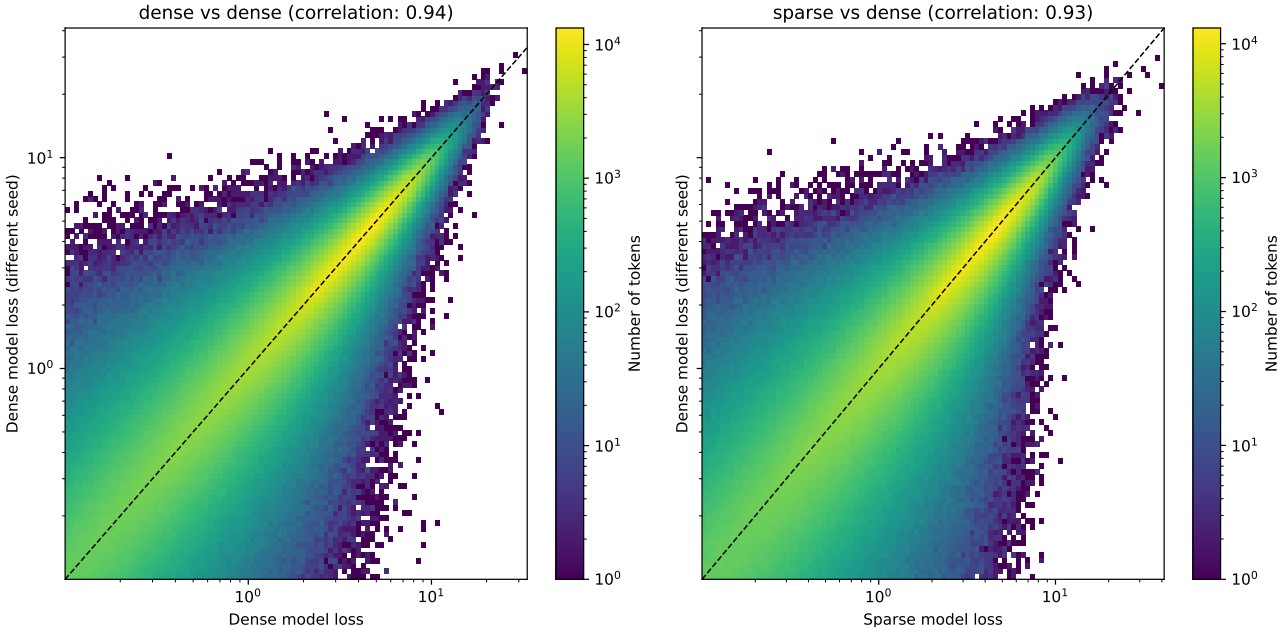

*Figure 33.* The loss of sparse and dense models on individual tokens is strongly correlated. The correlation is comparable to that between dense models trained with different seeds.

sparse GEMM kernels written to run on the CUDA cores instead of Tensor Cores. CUDA cores have no tiling requirements and can thus efficiently operate on sparse data, with the caveat that they are much slower. We find that CUDA core GEMMs are more efficient than their Tensor Core counterparts at sub-1% sparsity levels.

**Option 2: Semi-structured (2:4) sparsity.** Sparse Tensor Cores accelerate GEMMs when one operand satisfies a 2:4 pattern (at minimum two zeros in every contiguous group of four elements along the reduction dimension), with a *maximum* theoretical $2\times$ math-throughput speedup and reduced memory traffic. We experiment with 2:4 semi-structured sparsity and find that it shows some improvement in throughput, especially in regimes of less sparsity (i.e., 1% to 50%). However, the improvement is generally less than the theoretical 2x. Fusing prune and compress into the matmul is challenging, so to minimize memory bandwidth usage, we would ideally fuse the 2:4 prune and compress for both the forward and backward (transposed) layouts into the top-$k$ kernel.

**TopK kernels.** We find that significant time is spent on AbsTopK operations, both within the forward passes (enforcing activation sparsity) as well as during the training steps (enforcing weight sparsity). PyTorch's naive TopK implementation uses a radix sort operation, which is slow. We find large speedups by binary searching over a threshold to find k, which requires very few memory writes. This speedup is most apparent during activation sparsity, as the

small rows can be fit independently in different warps, allowing for massive parallelism across the batch dimension. This type of implementation can be sped up even more; we are excited about approaches for approximate top-$k$,[16] as the model does not seem especially sensitive to this during training, as long as the final few steps are done exactly.

Although these kernels would provide large speedups in theory, our current optimization stack is unsuited to using them during training. Significant further optimization research is needed to allow the use of sparse kernels everywhere, as well as sparse representations of all model parameters (to save memory). Early on in the project we experimented with various optimization changes to enable the kernel changes, but many of these systems changes would require architecting our codebase in a manner not conducive to rapid iteration, and so we ultimately decided to focus on research velocity for this work. Without additional research on systems approaches, we find it unlikely that we can drastically scale up our models, but we believe all of the problems outlined below to be tractable.

$dW$ **computation.** While the forward pass matmul and $dx$ matmul are both a sparse weight times a dense activation matrix, the $dW$ computation requires taking the product of

---

[16]It's worth being careful about approximate top-$k$. In some early experiments, we ran into some optimization degradation due to approximate top-$k$ with Key et al. (2024), because the buckets would line up with rows of the matrix, thereby inducing unintended structure.

two dense matrices. Thus, the $dW$ computation will become the limiting factor asymptotically. Importantly, only $L_0$ elements of $dW$ will actually be used in the computation, but those elements are chosen by top-$k$ of $W + dW$ (assuming we use SGD; we address Adam in the next section). While the memory usage is more straightforward to fix (we can fuse a TopK into the matmul kernel), the compute usage is a greater challenge. We believe it may be possible to approximate the top-$k$ of $W + dW$ operation, or amortize it out over many steps, similar to Evci et al. (2021).

**Adam moments.** One other source of dense memory and compute usage is the Adam moments. In early experiments, we explored pruning the Adam moments to the $m \cdot L_0$ most important entries, for various values of $m$ and measures of moment importance, but always found this to be a non-trivial optimization hit at reasonable $m$. There are several approaches we could take: we could use an optimizer like SGD or signSGD (Bernstein et al., 2018); we could accept the cost of Adam moment pruning, if the systems win overall is worth it; we could use some hybrid of sparse-moment Adam with a memory-efficient Adam-like optimizer Adafactor (Shazeer & Stern, 2018).

**Annealing.** We currently find that annealing our models from dense to sparse over a large amount of training helps substantially with optimization. If we were to significantly scale up, this would be a massive cost. We have also experimented with annealing schedules which decay rapidly over the first few training steps and still achieve similar final test loss.

## C. Binarizing feature magnitudes

When an SAE feature activates, it can take on a very wide range of different possible strengths. Therefore, to fully understand what a feature is doing, it would be insufficient to merely show that it activates if and only if a certain concept is present; we'd also have to either explain why it activated to the extent it did, or claim that the magnitude is of no significance (Chan et al., 2022).

Therefore, we binarized the model by inserting a step function at every node location, and then measuring the mean task loss.

**Binarization algorithm.** We construct an activation function $\psi_{t,\ell,r}$ such that $\psi_1$ is the identity function and $\psi_0$ is a step function

$$\psi_0(x) = \begin{cases} \ell & \text{if } x < (\ell + r)/2 \\ r & \text{otherwise,} \end{cases}$$

so that we can interpolate between the two, and such that

$\psi_t(\ell) = \ell$ and $\psi_t(r) = r$, by mixing the identity function with an appropriately-shifted sigmoid of temperature $t$. Throughout training, we anneal $t$ as $(1 - \text{progress})^5$.

To initialize the parameters $\ell$ and $r$ for each node, we iterate over each node and search over possible choices to find the choice that results in the least drop in performance. Specifically, we try thresholds $\frac{1}{4}$, $\frac{1}{2}$, and $\frac{3}{4}$ of the way in between the max and min observed activations, and take the mean of the subset of activations above and below the threshold for $\ell$ and $r$.

**Results.** We find that we can often binarize tasks, and that while the result is very noisy, there is generally a trend towards greater binarizability as total params increases while $L_0$ is held constant (Figure 34). However, unlike the circuit size metric, where decreasing $L_0$ improves interpretability, we generally find that *increasing* $L_0$ at a constant number of total params improves binarizability. All baseline task losses start out the same, because we prune to a fixed target task loss.

Anecdotally from qualitative exploration, some of our features do not seem binarizable, but still nonetheless seem to be understandable — for instance, some nodes, such as the one in `bracket_counting`, take on 3 distinct semantic values (for outside a list, inside a singly nested list, or inside a doubly nested list); other nodes, especially attention key channels, grow continuously throughout the context, in order to pay more attention to recent tokens.

## D. Smooth Approximations to L0 Norm

Previous work has discovered several techniques for enforcing sparsity in language models. In particular, Louizos et al. (2018) found success using a differentiable estimator of the $L_0$ norm based on the HardConcrete distribution. In this work, we benchmark this method against our TopK variant on a toy language modeling task.

In order to improve performance, we make two main modifications to the technique outlined in Louizos et al. (2018). Namely,

- **Initialization**: Louizos et al. (2018) opts to initialize all parameters to roughly the same value. Instead, we sample original parameter values from a scaled Bernoulli distribution, with a fixed initial sparsity.

- **Sparsity Floor**: Often, especially at high levels of sparsity regularization, the sparse model learns to deactivate all of its weights, stopping the model from learning further. To abate this, as well as provide more control over the final sparsity, we clip the sparsity penalty at a fixed minimum.

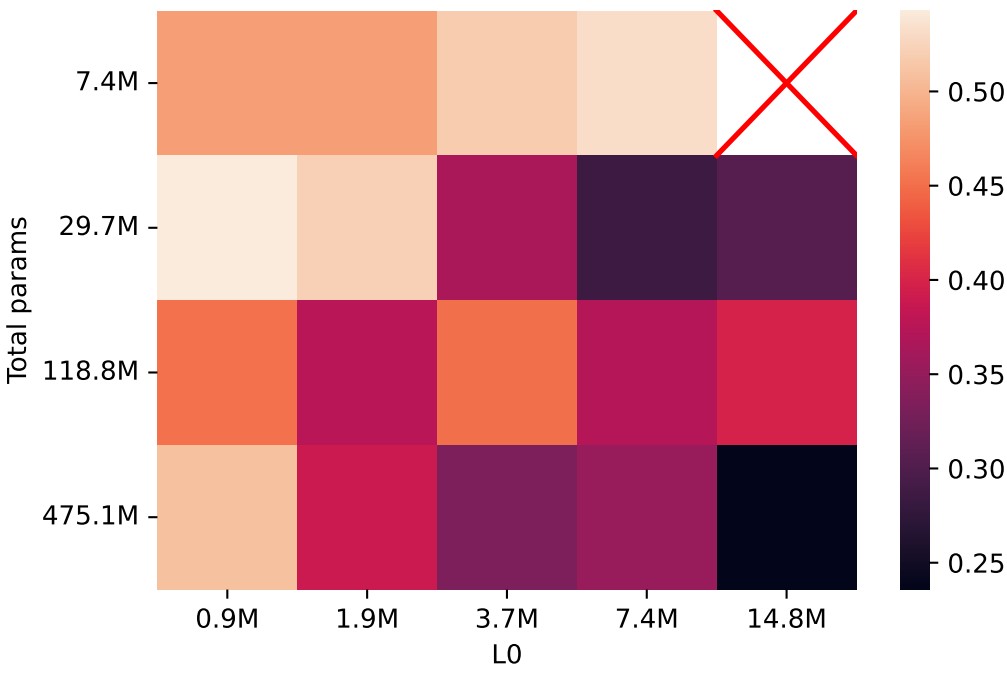

*Figure 34.* Task loss after binarizing all features in the circuit, average across all tasks. No data is available for $L_0 >$ total params for obvious reasons.

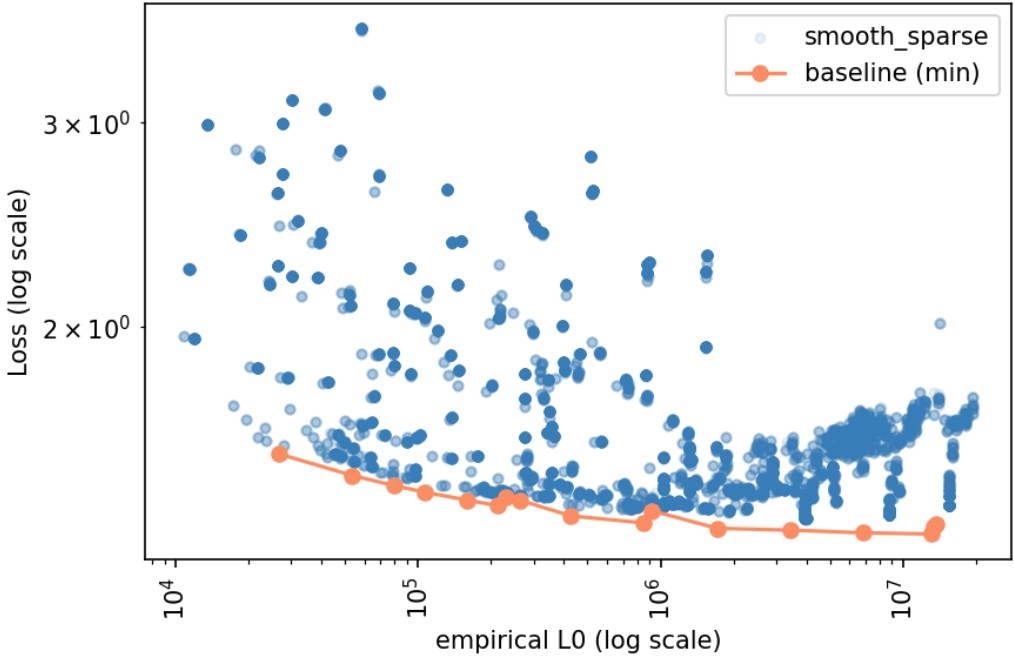

*Figure 35.* Comparing our method (baseline) to that of Louizos et al. (2018).

We find that the technique performs consistently worse than TopK, with slightly higher loss across all sparsity levels (Figure 35). This aligns with previous work, such as Gale et al. (2019), which benchmarked these techniques on Neural Machine Translation.

We also attempted to use a this technique for pruning, similar to Cao et al. (2021), and were unable to beat our baseline.

# E. Validating circuit hypothesis with mean ablations

Once we believe we've found a circuit inside our sparse model, we need to ensure that the circuit is actually reflective of the internal behavior of the model. Without faithfulness, explanations might look plausible but do not necessarily accurately reflect the model's underlying computation (Bolukbasi et al., 2021; Makelov et al., 2023; Friedman et al., 2024). Thus, in order to draw safety guarantees from interpretability, we must have some procedure for validating that our explanation is actually faithful.

Our work substantially improves on the state of the art for circuit faithfulness with very granular nodes, but it is still by no means maximally faithful.

We draw heavily on Causal Scrubbing (Chan et al., 2022) for inspiration. Unfortunately, our circuits are not fully faithful by the standards of causal scrubbing. Causal scrubbing has two components:

1. Any two node values that we claim to be semantically identical should be interchangeable. For example, if we claim that a neuron value is above $x$ if and only if the token is a variable containing a string, then it must be the case that changing the node to any in-distribution value greater than $x$ does not impact performance.

2. Any node we claim to be irrelevant must be substitutable with any value of the node drawn from the distribution of values observed during pretraining.

As we see in Appendix C, while our models satisfy condition 1 for some tasks, they are by no means uniformly able to do so. On this front, our method offers some signs of life, but does not perform well.

As for condition 2, using just the mean is a strictly weaker version of condition 2. In early results, we found that even just condition 2 of Causal Scrubbing alone was much harder to satisfy than mean ablation. However, we claim that the loss in faithfulness from using mean ablations instead of the full pretraining distribution is not so bad.

Chan et al. (2022) claim that mean ablation is suboptimal because it (a) can take the model out-of-distribution in an unprincipled manner, (b) can have unpredictable effects on

measured performance, and (c) remove variation that your model might depend on for performance. However, (a) only matters if your circuit actually depends on the irrelevant nodes, and even when it does, moving off the manifold of plausible activations should harm performance much more often than it helps; similarly, (b) plausibly increases variance and hurts the interpretability score on average, but seems unlikely to overestimate interpretability; to reduce (but not eliminate) the probability of (c) being a major issue, we also verify that ablating parts of the network that we claim to be relevant does indeed destroy performance (Figure 32).

Importantly, mean ablations leads to much more *complete* circuits than activation patching (Heimersheim & Nanda, 2024). If there are parts of the circuit that are critical for performing the task, but for which activations differ across prompt pairs rather than within each pair, then activation patching will fail to notice it. Concretely, for example, since each pair of prompts in set_or_string uses a different variable name, but the variable name is the same within each pair, activation patching will completely ignore the part of the circuit that copies the name of the variable to the final token. In early experiments, we found that activation patching was substantially easier to get good scores on, but led to circuits which were qualitatively unsatisfying.

We are excited for future work that further pushes the frontiers of circuit faithfulness to full causal scrubbing and beyond.

# F. Details for qualitative results

## F.1. Constant queries

In single_double_quote, the query channel appears to be a constant, and not data-dependent. To verify that this is a valid simplification, we show that setting it to a constant value across the entire pretraining distribution increases loss by 2.6e-5 nats per token, which is small compared to the pretraining loss hit of 1.47e-4 nats per token when zeroing the Q.

## F.2. Rescaling Ablations and Redundancy

For the circuit outlined in Section 3.2.2, to obtain the complete description, we found that rescaling a small number of nodes was helpful to remove redundant components of the model. However, this intervention is quite powerful when applied generally, and can "hide" superposition within the model.

We take care to ensure that we only perform rescaling interventions when the model computation are truly redundant, and are not hiding superposition present within the model. For the bracket_counting circuit, we rescale two activation scalars: 4.attn.resid_delta idx 1079

and 2.attn.resid_delta idx 1249.

For residual stream channel 1079, the circuit uses it in its final MLP layer, in order to compute the correct output logits. If this final MLP is ablated, loss suffers. However, this last layer can be ablated if one rescales 4.attn.resid_delta idx 1079 directly. This is equivalent to the logit_scale and logit_bias transformations used by pruning to recalibrate the output logits of the pruned circuits. We also find that by linearly replacing the unpruned outputs of the MLP is able to recover most of the loss, when compared to a baseline of zero ablation (5.8e-5 nats per token vs 2.69e-4).

We also find 3.attn to be implementing highly interpretable computation, essentially the "same circuit" as 2.attn in one attention head (head 85), and copying the 2.attn.resid_delta idx 1249 in another (1249). We ablate it mostly for simplicity.

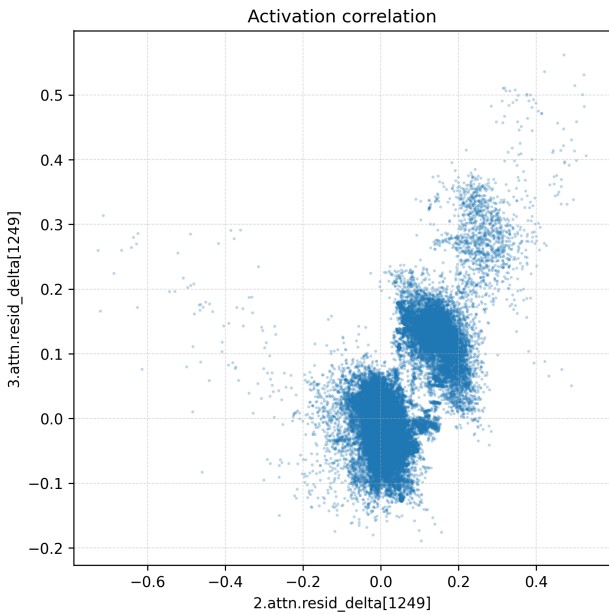

*Figure 36.* Activation of 2.attn.resid_delta idx 1249 vs 3.attn.resid_delta idx 1249 across pretraining.

For 2.attn.resid_delta idx 1249, it appears that within the `bracket_counting` circuit, residual channel 1249 is written to by attention layer 3 as well as attention layer 2. The activation patterns of 2.attn.resid_delta idx 1249 and 3.attn.resid_delta idx 1249 look highly correlated, both across the pretraining distribution, and the `bracket_counting` task distribution Figure 36. Thus, we suspect that the model is using 3.attn in order to amplify the activation from 2.attn. To verify this, we intervene on 3.attn.resid_delta idx 1249, replacing its activation with a linear function of 2.attn.resid_delta idx 1249, and compute pretraining loss. When compared to a baseline of zero ablation of the node (a 4e-3 nat per token loss hit), the linear replacement suffers a comparatively negligible loss hit of 7e-4 nats per token. Thus, this channel is unlikely to be in cross-layer superposition, and 3.attn is redundant at index 1249. To simplify the circuit description, we ablate 3.attn, and rescale 2.attn.resid_delta idx 1249 correspondingly.

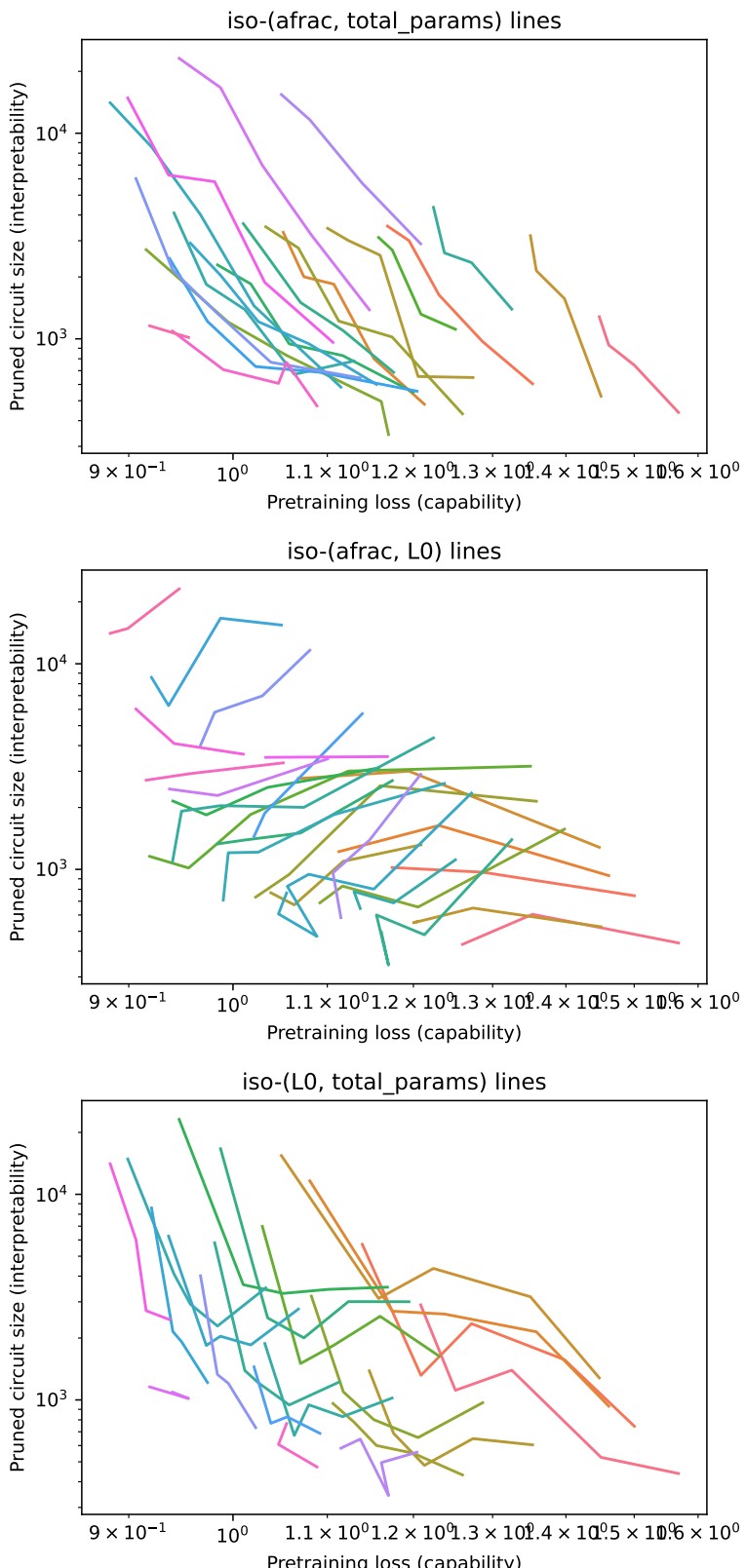

*Figure 37.* Contours varying one of $\{L_0,$ total parameters, activation sparsity$\}$ and holding the other two constant. L0 moves along the frontier. The first plot varies L0, the second varies total parameters, and the third varies activation sparsity. Total parameters pushes the frontier. Increasing activation sparsity initially helps but eventually becomes Pareto dominated.

```
=== 2.mlp.act_in[575] ===
  [5%] target -2.716184 (5.000%)
    Sample 1: activation -2.716183 (Δ +0.000001, |Δ| 0.000001)
        th_first_search(graph, start, end):
            # Initialize a queue and add the start node to it
            queue = deque([start])
            # Initialize a dictionary to keep track of names and distances
            distances = {start: 0}

            while queue:

    Sample 2: activation -2.716183 (Δ +0.000001, |Δ| 0.000001)
    _schedule, critical_sprinklers)
    for row in modified_schedule:
        print(row)

        print("Number of removed sprinklers:", deleted_sprinklers)<|endoftext|>def merge_data(row

    Sample 3: activation -2.716186 (Δ -0.000002, |Δ| 0.000002)
        the list unevenly, since each while loop can increment the i by 1 or more
            while i < len(loss) - 1:

                # increment the i if the cost of oversized boxes

  [95%] target 5.041825 (95.000%)
    Sample 1: activation 5.041825 (Δ +0.000000, |Δ| 0.000000)
    e+00",
        "-1.385742e+00", "1.165090e+01", "-8.964600e+00", "5.980754e+00",
        "-1.635742e

    Sample 2: activation 5.041822 (Δ -0.000002, |Δ| 0.000002)
    (polynomial_lfsr1, seed_lfsr1)
            self.lfsr2 = LFSR(polynomial_lfsr2, seed_lfsr2)

        def next_byte(self):
            by

    Sample 3: activation 5.041818 (Δ -0.000007, |Δ| 0.000007)
    .5

    # Define waypoints
    waypoints = [("GPS Point 1", 38.8977, 77.0365),
            ("GPS Point 2", 38.8895, 77.0

=== 1.attn.act_in[713] ===
  [5%] target -0.079470 (5.000%)
    Sample 1: activation -0.079459 (Δ +0.000011, |Δ| 0.000011)
    )
        return (newSkeleton.newElement(nodes=[n0, na, nc], parents=[element]),
                newSkeleton.newElement(nodes=[n1, nb,

    Sample 2: activation -0.079457 (Δ +0.000013, |Δ| 0.000013)
    (total_time, curr_time + d)
        return total_time

    # Test Case
    print(schedule_deliveries([2, 3,

    Sample 3: activation -0.079456 (Δ +0.000015, |Δ| 0.000015)
    otal_clear : y, 3:] = 1
                else:
                    a.map[x - a.total_clear:

  [95%] target 1.178244 (95.000%)
    Sample 1: activation 1.178240 (Δ -0.000003, |Δ| 0.000003)
    self.frm = frm

        def in_bound(size, point):
            return 0 <= point.y < size and 0 <= point.x < size

        def is_walkable(maze, point):
            return maze[point.y][point.x] == '.'

        def

    Sample 2: activation 1.178266 (Δ +0.000022, |Δ| 0.000022)
    +dx, y1 + t2*dy)
            ]
                return [pt for pt in intersections if 0 <= pt[0] <= 1 and 0 <= pt[1] <= 1]

        def area_of_intersection():

    Sample 3: activation 1.178200 (Δ -0.000044, |Δ| 0.000044)
    = len(maze[0])

        def is_valid(self, x, y):
            return 0 <= x < self.rows and 0 <= y < self.cols and self.maze[x][y] == 0

        def solve_maze(self, sx, sy, dx
```

```
=== 1.mlp.act_in[156] ===
  [5%] target -0.750229 (5.000%)
    Sample 1: activation -0.750229 (Δ -0.000001, |Δ| 0.000001)
            fee_list.append(node)

        return fee_list

    def build_spider_nets(node_list):
        spider_net_list = node_list[:]

        for node in node_list:
            children = node.generate_children()
            for child in

    Sample 2: activation -0.750231 (Δ -0.000002, |Δ| 0.000002)
            successors = self.problem.getSuccessors(current_node[0])
            for successor in successors:
                node = (successor[0], current_node, successor[1], current_node[3]+successor[2])
                if

    Sample 3: activation -0.750226 (Δ -0.000003, |Δ| 0.000003)
    is not in the list of processed nodes or queue,
            # add it to both.
            if next_node not in sysnodes and next_node not in queue:
                queue.append(next_node)
                sysnodes.append(next_node)

    # Print the list of proces

  [95%] target 0.000000 (95.000%)
    Sample 1: activation 0.000000 (Δ +0.000000, |Δ| 0.000000)
    ], scores['1307'][2], scores['1307'][3],
            "1308": (scores['1308'][0],

    Sample 2: activation 0.000000 (Δ +0.000000, |Δ| 0.000000)
    ])
        edges.append([36, 73])
        edges.append([36, 74])
        edges.append([37, 75])
        edges.append([37, 76])
        edges

    Sample 3: activation 0.000000 (Δ +0.000000, |Δ| 0.000000)
    closed_list):
                continue

                # Create the f, g, and h values
                child.g = current_node.g + 1
                child.h = ((c

=== 1.mlp.act_in[467] ===
  [5%] target 0.000000 (5.000%)
    Sample 1: activation 0.000000 (Δ +0.000000, |Δ| 0.000000)
    ], scores['1307'][2], scores['1307'][3],
            "1308": (scores['

    Sample 2: activation 0.000000 (Δ +0.000000, |Δ| 0.000000)
    ])
        edges.append([36, 73])
        edges.append([36, 74])
        edges.append([37, 75])

    Sample 3: activation 0.000000 (Δ +0.000000, |Δ| 0.000000)
    closed_list):
                continue

                # Create the f, g, and h values
                child.g = current_node.g + 1

  [95%] target 1.305829 (95.000%)
    Sample 1: activation 1.305828 (Δ -0.000000, |Δ| 0.000000)
    jacency list loaded successfully")
        return road_adj_dict

    def initialize_random_weights(road_adj_dict):
        weights_dict = {}
        for src in road_adj_dict:
            weights_dict[src] = {dest: math

    Sample 2: activation 1.305829 (Δ +0.000001, |Δ| 0.000001)
    ces(K, N, M, A, algorithm):
        def min_max_algorithm():
            allocated_resources = []
            unallocated_resources = M
            allocation_diff = []

            for i in range

    Sample 3: activation 1.305830 (Δ +0.000001, |Δ| 0.000001)
    as plt
    import random

    class Graph:
        def __init__(self, nodes, edges):
            self.nodes = nodes
            self.edges = edges

        def add_edges(graph, nodes_to_
```

*Figure 38.* A screenshot of the activation patterns of some random (post RMSNorm) residual stream features from one of our models. Notably, we do a pretraining pruning step to remove dead nodes which do not before we select random nodes. For each node, random documents from the top and bottom 5 percentile of activation are shown. Random nodes are somewhat interpretable, especially considering that our models are unlikely to represent extremely complex concepts.

bottom 0.1%

```
"."],".",".",".",".",".",".","S",".","."],
     [".",".",".","S","S",".",".",".",".",".",".","."],
     [".","S",".",".",".",".","S",".","."]
```

top 0.1%

```
                              # log results to a file
                              with open('runtime.txt', 'w')
as f:
                                  f.write(f'Shortest
Path Algorithms Execution Time\n')
                                      f.write(f'-----------
---------------------------\n
```

bottom 1.0%

```
"24695", "24695", "24695", "24695", "24695",
"24695", "24695",
                    "24695", "24695", "24695",
"24695", "24695", "24695", "24695", "24695",
"24695",
                    "24695", "24695", "24695",
"
```

top 1.0%

```
00000")
data = re.sub(r'[000-000]', "", data)

# Extracting and formatting comments
comments = re.findall(r'<\w+> (
```

bottom 10.0%

```
}")
    for i in range(nVAR):
        print(f"X{i + 1} =
{current_best.X[i]}")

branch and bound("IN02", solver_type="BIG_M",
problem type="PURE", MAX NODES
```

top 10.0%

```
, 490]],
              'S3C3': [f'CS{i}' for i in [331,
401, 413, 415, 419, 490]],
              'S4C1': [f'CS{i}' for i in [325,
331, 335, 345, 400, 404, 412, 416
```

bottom 50.0%

```
1
    return dependencies

def get independent node(dependencies):
    for node, dependency_count in
dependencies.items():
        if dependency count == 0:
            return node
    return None
```

top 50.0%

```
(1, m):
        num sets = len(dice_sets)
        dice sets =
np.concatenate((np.repeat(dice sets, num dice,
axis=0), np.tile(dice_indecies, (num_sets
```

*Figure 39.* A screenshot from our circuit visualizer showing documents with various percentiles of activations, drawn from the entire pretraining distribution (and not just the single_double_quote task) for the 10.attn.resid_delta.83 node from Figure 4. Even on the pretraining distribution, the activations are surprisingly monosemantic—it activates positively inside double quote strings, negatively inside single quote strings, and near zero outside of strings. This node is cherrypicked; not all nodes are this monosemantic.

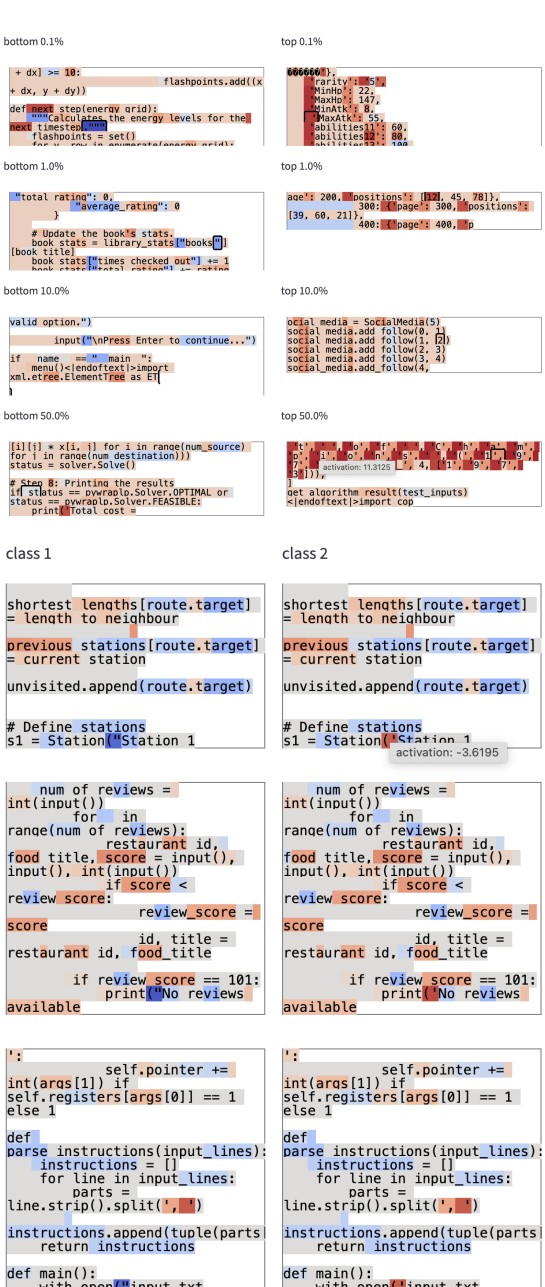

*Figure 40.* A screenshot of visualized activations for a quote type classifier layer 3 attention input channel of a bridged sparse model used for constructing interpretable perturbations. Most positive and negative pretraining activations are on top and paired task activations are below.

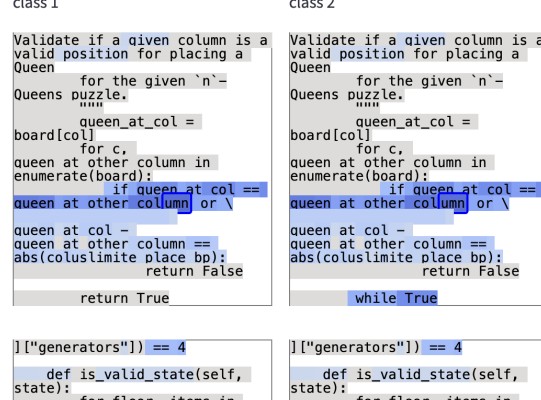

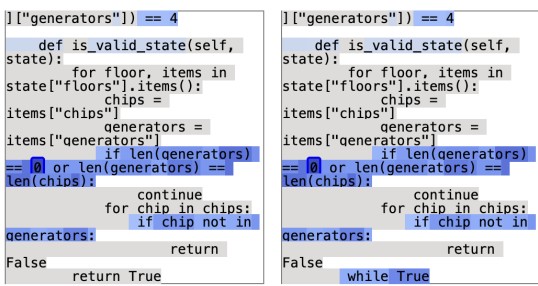

*Figure 41.* A screenshot of visualized activations for a "current line begins with if/while/except" layer 3 MLP input channel in a bridged sparse model used for constructing interpretable perturbations. Most negative pretraining activations on top and paired task activations below.

# G. Contributions

**Leo Gao** set the research direction and led the project. Leo designed and implemented the sparse model training code-base and the pruning codebase. Leo studied scaling, optimization, architecture, pruning, bridges, and feature binarization. Leo worked on the pretraining systems and kernels. Leo created several of the pruning tasks. Leo contributed to the visualizer. Leo contributed substantially to the writing of the paper text. Leo provided technical mentorship for Achyuta through the duration of the project.

**Achyuta Rajaram** studied pretraining architecture, optimization, and circuit pruning. Achyuta improved the pruning algorithm. Achyuta performed the qualitative analysis for the examples in the paper. Achyuta created many of the pruning tasks. Achyuta experimented with alternatives to top-k for weight sparsity. Achyuta contributed substantially to the writing of the paper text.

**Jacob Coxon** studied optimization and circuit pruning. Jacob did initial explorations into $L_0$ annealing and attribution-based pruning. Jacob designed an initial version of the dataset. Jacob created several of the pruning tasks. Jacob implemented the circuit visualizer.

**Soham V. Govande** implemented an optimized CARBS implementation and contributed to the kernels and systems analysis.

**Bowen Baker** managed Leo for the first portion of the project.

**Dan Mossing** studied bridges and gave day-to-day technical feedback. Dan contributed substantially to the writing of the paper text. Dan managed Jacob and Achyuta through the duration of the project, and Leo for the latter portion of the project.

