# OpenReview forum: "Weight-sparse transformers have interpretable circuits"
_ICML.cc/2026/Conference — ICML 2026 regular_

### Official Review · Reviewer_jxVY · 2026-03-09

**Soundness:** 3
**Presentation:** 2
**Significance:** 3
**Originality:** 4
**Overall Recommendation:** 5
**Confidence:** 4

**Summary:**

This paper studies the scaling behavior and architectural design of a transformer-based model under pruning and efficiency constraints. The authors explore how model capacity and architectural components interact with training objectives and investigate the relationship between pruning strategies and model performance. The paper aims to provide insights into scaling laws and architectural efficiency in language models.

**Compliance With Llm Reviewing Policy:**

Affirmed.

**Final Justification:**

The rebuttal addressed my concerns. Reinforced my prior assessment

**Key Questions For Authors:**

Have the authors conducted ablation studies on the loss function? If so, how sensitive are the results to different loss formulations?


Why is 0.15 chosen as the target loss in the pruning experiments? Is this value empirically chosen, or does it correspond to a specific benchmark or performance threshold?

What exactly is the bigram table mentioned in the appendix architecture, and how is it used in the model?

**Limitations:**

yes

**Strengths And Weaknesses:**

Strength：

The paper addresses an important problem related to model scaling and efficiency, which is highly relevant for modern large language models.

The exploration of pruning and scaling relationships could potentially provide useful insights for designing more efficient models.

The paper includes several experimental results and architectural discussions that could be valuable for future research.




Weakness：

Discussions in the details:

The paper suggests that token semantics are understood at around 10 layers. How do the authors ensure or verify this observation? If semantic understanding emerges at this depth, what roles do the preceding shallow layers play in the downstream tasks?

Many of the explanations in the paper appear to be presented on a case-by-case basis. However, the paper claims to derive insights about scaling laws. The connection between individual experimental observations and a unified scaling-law interpretation is not sufficiently clear.

Other experiment details are in the question part.

---

> ### Author Rebuttal · Authors · 2026-03-31
>
> Thank you for your thoughtful review and insightful comments. We are excited about this work as a scalable method to train interpretable language models. Our results show evidence that not only can we learn qualitatively interpretable circuits for a variety of tasks (Figures 4,5,6), but we can predictably improve model performance and interpretability by scaling compute (Figure 3).
> To address your specific questions:
>
>
> 1. For the purposes for this work, we studied autoregressive transformer language models, and therefore use the standard NLL objective. We have not carefully ablated this loss function choice, but we expect similar techniques as the ones in the paper to allow stable training under other common loss functions as well.
>
>
> 2. For the pruning step, we choose 0.15 as a target loss as it empirically balances task performance and circuit size, yielding interpretable circuits which complete the tasks of interest. On our binary choice tasks, this corresponds to an accuracy of approximately 85%. This choice is somewhat arbitrary; Figure 2 shows that you can smoothly trade off pruned circuit size for loss, so other choices of target loss are valid.
>
>
> 3. The bigram table is an architectural choice that improves the performance of our sparse transformers (Figure 24) while preserving interpretability. In a standard transformer, residual connections allow the embedding and unembedding layers to encode bigram statistics. Our model cannot easily do this, as the embedding and unembedding are sparse matrices. We therefore ensemble the sparse transformer with an $n_{\text{vocab}} \times n_{\text{vocab}}$ lookup table, which assigns a distinct logit distribution to each input token, and sum their outputs to produce the final logits. This lookup table is left dense: unlike the other parameters in the model, we allow all of its entries to be nonzero.

---

> > ### Author Rebuttal · Reviewer_jxVY · 2026-04-01
> >
> > Thanks for the response. I'll maintain my positive score

---

### Official Review · Reviewer_eJCK · 2026-03-10

**Soundness:** 4
**Presentation:** 3
**Significance:** 3
**Originality:** 4
**Overall Recommendation:** 6
**Confidence:** 4

**Summary:**

This paper introduces a method for training sparse-weights (and sparse-activations) models from scratch, and trains a GPT-2-small-scale model on Python code. It finds that this model has interpretable circuits: because the model has sparse activations, no sparse dictionaries are needed to decompose its activations into features; because the weights are sparse, it is possible to directly observe how features / activations are manipulated and combined to create other features. The authors verify the sufficiency (and necessity) of these circuits via mean ablations, and use the circuits to predict model behavior on unseen inputs. The authors also propose and train "bridges", which map between the activations of sparse and dense models; these bridges allow for interventions in dense models based on the interpretable features of the sparse models.

**Compliance With Llm Reviewing Policy:**

Affirmed.

**Final Justification:**

I'm satisfied by the rebuttal, and will stick with my positive assessment.

**Key Questions For Authors:**

- Does this method impose any fundamental limitations on what sort of tasks a model can learn to perform? Or do the limitations essentially stem from the difficultly and expense of training these models?
- Relatedly, do you have any intuitions regarding which tasks can be learned satisfactorily by this sort of model? What are the barriers to training a sparse GPT-2 trained on more general web text data?
- What is the failure mode if such a sparse model fails to learn the task? You say that "We generally don’t find improvements of the capability-interpretability pareto frontier from increased token budget" as task loss decreases, while interpretability worsens. Is there a way to detect when this is happening, or to discern the optimal token budget
- To what extent will bridges learn faithful mappings between dense and sparse model activations? Past work on SAEs has stressed how e.g. the activation functions of SAEs correspond to assumptions about feature geometry that may not be fulfilled by the features of (dense) models. Does this similarly imply that these bridges will be limited in their ability to translate certain features from dense to sparse models? And (as asked above) does this suggest a more fundamental issue for sparse-activation models?

**Limitations:**

yes

**Strengths And Weaknesses:**

**Soundness**: In general, this paper seems sound. Interpreting features via max-activating examples, and verifying these interpretations by performing causal interventions and confirming that the ablation has the expected effect is pretty standard. I appreciate that the authors were also able to predict held-out behavior and test whole-circuit faithfulness (which happens more in component circuits than feature circuits). I found the bridge experiments somewhat less thorough and compelling, but they seem acceptable nonetheless. Overall, this paper gives the impression that the authors have considered many of the most common pitfalls in mechanistic interpretability (and many of the less common ones as well).

**Presentation**: Considering the complexity of the topic, I think that this paper is pretty readable. Figure 4 is quite complex, but I appreciate that you can see the entire circuit. Figure 1 seems comparatively information-sparse; more could be done with this space. There are two cases where there is a space missing in the phrase ".See" (L157, L273 (footnote))

**Significance**: Finding circuits of interpretable features has been a significant goal of the mechanistic interpretability community, and I think that anyone for whom that goal is important will find this paper interesting. My understanding is that the main barrier to impact for this paper is the computational expense of training this sort of model; this sort of model is expensive to train even at relatively small sizes, and on relatively simple tasks. That said, the authors make a valiant effort to highlight research / engineering directions that could make training cheaper. I also think that this technique would be more significant if training code were released, given the complexity of the training process.

**Originality**: This work is, as I see it, quite original. It bridges two approaches in mechanistic interpretability - sparse decomposition of activations into features, and sparse decomposition (or pruning) of weights - into one approach for creating weight- and activation-sparse models. I do think it's worth citing even earlier work that did interpretability attempting to totally reverse-engineer smaller models, from recent work like Nanda et al. / Zhong et al. (2023) to older connectionist work like Rumelhart. But this paper is in any case novel, and does a better job at discussing related work that many papers in the field.

---

> ### Author Rebuttal · Authors · 2026-03-31
>
> Thank you for your insightful comments. We are excited to scale this technique further and expect to uncover additional differences between dense and weight-sparse transformers as we do.
>
> To address some of your specific questions:
>
> In the settings we studied, we find preliminary evidence that bridges seem to learn faithful mappings between dense and sparse model activations by performing causal studies on two of our tasks (Figure 9). However, more research is needed, since the current bridge design inherits many of the limitations and assumptions of SAEs, by only being able to support linear transformations. As we believe that these limitations exist, we are excited about future work that finds concrete examples of what features bridges cannot well represent, as well as better bridge architectures which relax them.
>
> However, we do not currently believe that weight sparsity imposes any fundamental limitations on what sort of algorithms a model can implement. From a theoretical standpoint, consider two models with the same $L_0$, but one which is much wider than the other. The wider model can effectively reduce its $d_{model}$ by restricting its weight matrices to only use a subset of all columns and rows, thus simulating the smaller one.
>
> Obviously, these kinds of theoretically equivalent solutions are never learned. In practice, we actually empirically find that the models become both more performant and more interpretable if you scale them up while fixing $L_0$ (Figure 3)! Thus, the important question is whether the inductive bias of weight sparsity causes sparse models to learn a different set of algorithms than dense ones. We explore this in Figure 33, finding some preliminary evidence that the token-level loss of sparse and dense models are highly correlated.
>
> When we look at the tokens which have the highest loss difference between the sparse and dense transformers, there exist some weak patterns which look like interpretable phenomena. One example is that we believe that our models currently have some difficulty learning induction heads; it's unclear how important or fundamental this limitation is, as it could disappear at scale as the model learns alternative algorithms.
>
> Overall, we do not believe that there are any deep fundamental limitations to training a sparse GPT-2 on more general web text data. Most of the work ahead lies in engineering and systems for large-scale distributed training (see Appendix B), further optimization research to improve the interpretability-capability frontier, and dense sweeps to understand the optimal way to scale all relevant variables (L0, token budget, compute, model size, etc).

---

> > ### Author Rebuttal · Reviewer_eJCK · 2026-04-01
> >
> > Thanks for the rebuttal; I'm satisfied by it, and will keep my (high) score.

---

### Official Review · Reviewer_67CW · 2026-03-12

**Soundness:** 2
**Presentation:** 2
**Significance:** 3
**Originality:** 2
**Overall Recommendation:** 4
**Confidence:** 4

**Summary:**

The paper presents training weight sparse transformers as a viable method to isolate and interpret circuits in transformers trained on a specific task, as an alternate to the existing methods such as Sparse Autoencoders which abstracts away complex computations. The method involves training the transformers with sparser weights where majority of the weights are zero, constraining the neurons to be able to only read from and write to a few residual stream channels, preventing the concepts and computation to be distributed across multiple neurons, incentivizing the neuronal basis to be naturally interpretable.

The method is evaluated on a suite of python binary prediction tasks such as string closing, bracket nesting detection and variable type tracking, by pruning the model down to the smallest circuit that still performs well and interpreting them. The paper also measures interpretability operationally as the size of the circuit, used to compare dense and sparse weight transformers. The paper finds that the sparse weight models have consistently smaller circuits while matching the loss of the dense baselines.

The other contribution is a bridge based approach for interpreting existing dense models. The authors train a sparse model along linear encoder/decoder "bridges" that maps activations between dense model and sparse model at each sublayer. This lets them use the sparse model as an interpretable proxy.

**Compliance With Llm Reviewing Policy:**

Affirmed.

**Key Questions For Authors:**

Refer weaknesses

**Limitations:**

yes

**Strengths And Weaknesses:**

Strengths :
S1. First comprehensive study to evaluate weight-sparse training as a method to train models that are interpretable by nature unlike existing methods that work to interpret / disentangle dense uninterpretable models as an additional step.
S2. The qualitative examples are concrete and thorough in demonstrating and explaining how the models perform the specific tasks
S3. Provides a thorough scaling evidence, showing the capability-interpretability frontier, and finds that increasing model size can improve the frontier, while changing sparsity budget moves along it.

Weaknesses :
W1. The bridges seem to be under validated leading me to believe that it is a costly alternate to interpreting existing dense models, where the Sparse Autoencoders and its variants seem more easier and straight forward to work with at first glance.
W2. The practical inefficiency of requiring enormous compute as stated limits the method's reach and further work in this direction
W3. Anecdotal language such as "a researcher-day of work" seems a bit hand wavy and would benefit from a slightly well defined measure.

---

> ### Author Rebuttal · Authors · 2026-03-31
>
> Thank you for your review.
>
> We believe that, although our method is harder to use than SAEs, sparse circuits bring us closer to a full understanding of model internals. Sparse circuits enable deeper analysis of the structural connections between features, and therefore of the algorithms implemented by transformers. In general, we believe that this technique allows you to spend more compute (as well as some implementation complexity) in order to achieve more faithful and rigorous explanations than ones an SAE can provide on its own. We believe it would be extremely valuable to fully understand even relatively simple models, even if doing so required enormous compute.
>
> We agree that our use of anecdotal language, describing interpreting our circuits as "a researcher-day of work", is somewhat hand-wavy. The main reason for its inclusion is to ground our well-defined quantitative metric of pruned circuit size (See section 2.2), in something more relatable to the reader. We wish to capture the reality that the process of manually interpreting the pruned circuits took a moderate amount of manual labor, but was still quite doable. We intend to open-source all of the circuits found in this work to allow the community to verify these claims.

---

> > ### Author Rebuttal · Reviewer_67CW · 2026-04-02
> >
> > My concerns regarding benchmarking against transcoders, and clarification on bridges have not been addressed. I maintain my score

---

### Decision · Program_Chairs · 2026-04-30

**Decision:**

Accept (regular)

**Comment:**

This paper trains weight-sparse transformers from scratch such that neurons have only a few connections, producing circuits that are directly human-readable without post-hoc dictionary learning. The authors demonstrate interpretable circuits on Python code tasks, characterize the capability-interpretability scaling frontier, and propose "bridges" to port sparse-model interpretations to existing dense models.

Reviewer consensus is strongly positive (6/5/4). The method is praised for rigorous verification via mean ablations, held-out behavior prediction, and whole-circuit faithfulness tests (eJCK, jxVY). The scaling analysis of the capability-interpretability tradeoff is particularly valuable, providing quantitative guidance on sparsity levels that balance capability retention with interpretability gains (eJCK). The paper is well-written with clear presentation and thorough experimental methodology.

The bridge approach is acknowledged as preliminary and computationally expensive, and benchmarking against transcoders was not fully addressed (67CW). These are noted limitations but do not undermine the core contribution. All reviewers maintained positive scores after rebuttal. Among the most rigorous mechanistic interpretability contributions in this cycle, demonstrating a compelling alternative paradigm — training for interpretability rather than explaining post hoc.